# Adherence to the EAT-Lancet diet and incident depression and anxiety

Xujia Lu ®[1], Luying Wu[1], Liping Shao[1], Yulong Fan[1], Yalong Pei[1], Xinmei Lu[1], Yan Borné[2,3] & Chaofu Ke ®[1,3] ✉

High-quality diets have been increasingly acknowledged as a promising candidate to counter the growing prevalence of mental health disorders. This study aims to investigate the prospective associations of adhering to the EAT-Lancet reference diet with incident depression, anxiety and their co-occurrence in 180,446 UK Biobank participants. Degrees of adherence to the EAT-Lancet diet were translated into three different diet scores. Over 11.62 years of follow-up, participants in the highest adherence group of the Knuppel EAT-Lancet index showed lower risks of depression (hazard ratio: 0.806, 95% CI: 0.730–0.890), anxiety (0.818, 0.751–0.892) and their co-occurrence (0.756, 0.624–0.914), compared to the lowest adherence group. The corresponding hazard ratios (95% CIs) were 0.711 (0.627–0.806), 0.765 (0.687–0.852) and 0.659 (0.516–0.841) for the Stubbendorff EAT-Lancet index, and 0.844 (0.768–0.928), 0.825 (0.759–0.896) and 0.818 (0.682–0.981) for the Kesse-Guyot EAT-Lancet diet index. Our findings suggest that higher adherence to the EAT-Lancet diet is associated with lower risks of incident depression, anxiety and their co-occurrence.

Mental health disorders, the leading cause of avoidable suffering and premature mortality, are a growing global health crisis[1]. Mental health disorders affect almost 30% of individuals across their lifespan[2] and were the seventh leading cause of disability-adjusted life-years (DALYs), which accounted for 125 million DALYs in 2019[3]. Among the mental disorders, depression and anxiety are of particular concern as they ranked first and second contributors to DALYs in 2019[4,5]. Depression and anxiety increased the risks of incident cardiovascular disease[6], frailty and all-cause mortality[7], and growing evidence has shown the coexistence of depression and anxiety was associated with severer consequences than each disorder alone[8]. Moreover, the emergence of COVID-19 pandemic has made public mental health issues eminently crucial[9,10]. An additional 53.2 and 76.2 million cases of depression and anxiety globally were estimated due to the COVID-19 pandemic[9]. Considering the substantial impairment and immense health burden imposed by depression and anxiety, evaluating them together and identifying modifiable risk factors for the primary prevention of depression and anxiety constitutes an urgent public health priority.

According to the International Society for Nutritional Psychiatry Research, nutritional medicine was suggested to be considered mainstream in psychiatric practice with the support of research, education, policy, and health promotion[11,12]. Due to the essential role of nutrients in the neuroendocrine system, poor diet quality resulting in unbalanced nutrient intake is an important risk factor for the development of mental disorders, and thus diet has been suggested as a promising candidate as a public health target for the prevention and treatment of these illnesses[12,13]. When examining the association between diet and mental disorders, dietary pattern analysis surpasses analyses of single nutrients or foods as diet patterns could reflect the numerous and multifaceted combinations of nutrient and food consumption[14,15]. The associations of diet patterns (e.g., Mediterranean diet[16], Dietary Approaches to Stop Hypertension (DASH) diet[17] and plant-based diet[18,19]) with the risk of depression or anxiety have also been previously reported. However, these dietary patterns mainly focus on human health. In the context of the significant negative effects on the environment due to the current food production[20,21], the

[1]Department of Epidemiology and Biostatistics, School of Public Health, Suzhou Medical College of Soochow University, Suzhou, China. [2]Department of Clinical Sciences Malmö, Lund University, Malmö, Sweden. [3]These authors jointly supervised this work: Yan Borné, Chaofu Ke. ✉e-mail: cfke@suda.edu.cn

EAT-Lancet Commission proposed the EAT-Lancet reference diet in 2019 with the aim of increasing both planetary and human health[22]. The EAT-Lancet reference diet takes into account the multiple links between health, nutrition intake and the environment, and provides a set of recommendations for feeding the global population within planetary boundaries[23]. Overall, this dietary pattern is primarily plant based and emphasizes the intake of vegetables, fruits, whole grains and nuts. It consists of moderate amounts of seafood and poultry and considerably limits the intake of red meat, added sugar and saturated fat[22]. Existing observational studies have demonstrated inverse associations of the EAT-Lancet reference diet with multiple diseases and mortality[24–28]. However, evidence on the association between the EAT-Lancet reference diet and mental health is absent, and to the best of our knowledge, no studies have evaluated whether the EAT-Lancet dietary pattern is associated with the risk of developing depression and anxiety.

In this work, we aimed to investigate the prospective associations between adherence to the EAT-Lancet reference diet and new-onset depression, anxiety and co-occurrence of depression and anxiety in a large population-based cohort of UK adults. The results show that greater adherence to the EAT-Lancet diet is associated with lower risks of incident depression, anxiety and their co-occurrence. The associations are consistent among the 3 common yet differently quantified EAT-Lancet indexes. The findings suggest that the prevention of depression and anxiety might be achieved by adhering to this sustainable dietary pattern.

## Results

### Baseline characteristics
A total of 180,446 participants were included in the study, among whom the mean age (standard deviation, SD) was 56.2 (8.0) years at baseline, and 83,824 (46.45%) were male (Table 1). During a median (interquartile range, IQR) follow-up of 11.62 (11.00–12.43) years, 4548, 6026 and 1262 incident cases of depression, anxiety and their co-occurrence were identified, respectively. Participants in this study obtained between 5 and 14 points on the Knuppel EAT-Lancet index, with a median (IQR) of 11 (10–11) points, between 8 and 38 points on the Stubbendorff EAT-Lancet index, with a median (IQR) of 22 (20–25) points, or obtained between −174 and 530 points on the Kesse-Guyot EAT-Lancet index, with a median (IQR) of 44 (24–67), respectively. In terms of baseline characteristics according to categories of the Knuppel EAT-Lancet index, participants with higher EAT-Lancet diet scores were more likely to be women, slightly older, nonsmokers, more physically active, and had lower body mass index (BMI) and total energy intake. Baseline characteristics according to the Stubbendorff EAT-Lancet index or Kesse-Guyot EAT-Lancet index can be seen in Supplementary Tables 1 and 2.

### Associations of the Knuppel EAT-Lancet index (range, 0–14 points) with incident depression and anxiety
Restricted cubic splines (Fig. 1) showed that the EAT-Lancet index associated, in a linear dose-response manner, with risks of incident depression, anxiety and their co-occurrence (all *P* for non-linearity > 0.05). In the fully adjusted Cox regression models, the respective hazard ratios (HRs) and 95% confidence intervals (95% CIs) of incident depression, anxiety and their co-occurrence were 0.806 (0.730–0.890), 0.818 (0.751–0.892) and 0.756 (0.624–0.914) for the highest vs lowest groups of the EAT-Lancet diet index (Table 2). Each additional point of the EAT-Lancet diet score was associated with a 5.1% decreased risk of depression (HR: 0.949, 95% CI: 0.925–0.974), a 4.7% decreased risk of anxiety (HR: 0.953, 95% CI: 0.932–0.975), and a 6.3% decreased risk of co-occurrence of depression and anxiety (HR: 0.937, 95% CI: 0.892–0.985), respectively.

### Associations of the Stubbendorff EAT-Lancet index (range, 0–38 points) with incident depression and anxiety
Restricted cubic splines (Fig. 1) showed linear associations between EAT-Lancet index and depression, anxiety and co-occurrence of depression and anxiety (all *P* for non-linearity > 0.05). When comparing the highest with the lowest groups of adherence, the multivariable HR (95% CI) for incident depression was 0.711 (0.627-0.806), and the multivariable HR (95% CI) for incident anxiety was 0.765 (0.687–0.852), respectively (Table 3). When evaluating the EAT-Lancet index as a continuous variable (0–38 points), each incremental increase in points contributed to a 2.6% lower risk of incident depression (HR: 0.974, 95% CI: 0.966–0.982) and a 1.9% lower risk of incident anxiety (HR: 0.981, 95% CI: 0.974–0.988). Finally, when testing the association of the EAT-Lancet index with co-occurrence of depression and anxiety, the HRs (95% CIs) were 0.659 (0.516–0.841) for the highest adherence group and 0.971 (0.956–0.986) for 1-point increment in diet score.

### Associations of the Kesse-Guyot EAT-Lancet index with incident depression and anxiety
Restricted cubic splines (Fig. 1) showed linear associations between EAT-Lancet index and depression, anxiety and co-occurrence of depression and anxiety (all *P* for non-linearity > 0.05). After full adjustment, results from Cox regression models showed that compared with the lowest quintile (quintile 1), the HRs (95% CIs) of quintile 5 of EAT-Lancet index were 0.844 (0.768–0.928) for incident depression, 0.825 (0.759-0.896) for incident anxiety and 0.818 (0.682–0.981) for co-occurrence of depression and anxiety, respectively (Table 4). Each 100-points increase of the EAT-Lancet index was associated with a 13.6% decreased risk of depression (HR: 0.864, 95% CI: 0.795–0.938), a 15.8% decreased risk of anxiety (HR: 0.842, 95% CI: 0.784–0.905), and a 16.8% decreased risk of co-occurrence of depression and anxiety (HR: 0.832, 95% CI: 0.711–0.973).

### Predictive performance of the EAT-Lancet index
Table 5 shows that the net reclassification improvement (NRI) indexes (95% CIs) of adding the Stubbendorff EAT-lancet index to the reference model were 0.112 (0.082–0.138) for depression, 0.088 (0.062–0.116) for anxiety and 0.146 (0.058–0.202) for their co-occurrence, respectively. Likewise, adding the Kesse-Guyot EAT-Lancet index also significantly improved the risk reclassification for depression (NRI: 0.060, 95% CI: 0.026–0.092), anxiety (NRI: 0.044, 95% CI: 0.018–0.072) and their co-occurrence (NRI: 0.090, 95% CI: 0.010–0.136). Adding the Knuppel EAT-Lancet index significantly improved the risk reclassification for incident depression, with NRI (95% CI) of 0.038 (0.008–0.064). In addition, adding the Mediterranean diet index, the DASH diet index and the unhealthful plant-based diet index to the reference model brought significant improvement to the risk reclassification for incident depression, with respective NRIs (95% CIs) of 0.042 (0.012–0.060), 0.082 (0.052–0.112) and 0.112 (0.080–0.142). The corresponding NRI (95% CI) of adding the unhealthful plant-based diet index to the reference model was 0.084 (0.060–0.106) for incident anxiety and the NRIs (95% CIs) of adding the DASH diet index and the unhealthful plant-based diet index to the reference model were 0.078 (0.012–0.132) and 0.142 (0.088–0.190) for their co-occurrence, respectively.

### Sensitivity and subgroup analyses
We observed similar inverse associations between the EAT-Lancet index and incident depression, anxiety and their co-occurrence within further analyses, including excluding participants that only completed the online 24 h dietary recall questionnaire on one occasion (Supplementary Tables 3–5), using follow-up time which began at the time when the latest dietary assessment was completed (Supplementary Tables 6–8), and excluding depression, anxiety or co-occurrence cases

**Table 1 | Baseline characteristics of the study participants according to categories of the Knuppel EAT-Lancet index (range, 0–14 points) (N = 180,446)**

| Characteristics | Total | Categories of the Knuppel EAT-Lancet index | | | |
|---|---|---|---|---|---|
| | | ≤ 9 | = 10 | = 11 | ≥ 12 |
| N | 180446 | 34027 | 54919 | 58378 | 33122 |
| Age (years) | 56.16 ± 7.97 | 55.04 ± 8.18 | 56.08 ± 8.03 | 56.60 ± 7.87 | 56.67 ± 7.70 |
| Sex (male, %) | 83824 (46.45) | 20856 (61.68) | 28192 (51.63) | 24227 (41.70) | 10120 (30.72) |
| Ethnic (%) | | | | | |
| White | 172018 (95.33) | 32310 (94.95) | 52519 (95.63) | 55777 (95.54) | 31412 (94.84) |
| Mixed | 1057 (0.59) | 221 (0.65) | 312 (0.57) | 326 (0.56) | 198 (0.60) |
| Asian or Asian British | 2573 (1.43) | 401 (1.18) | 672 (1.22) | 813 (1.39) | 687 (2.07) |
| Black or Black British | 2278 (1.26) | 608 (1.79) | 686 (1.25) | 656 (1.12) | 328 (0.99) |
| Other | 1862 (1.03) | 351 (1.03) | 529 (0.96) | 599 (1.03) | 383 (1.16) |
| Townsend score (%) | | | | | |
| Above median | 90124 (49.95) | 17810 (52.34) | 26979 (49.13) | 28406 (48.66) | 16929 (51.11) |
| Others | 90322 (50.05) | 16217 (47.66) | 27940 (50.87) | 29972 (51.34) | 16193 (48.89) |
| Smoking status (%) | | | | | |
| Never smoked | 103269 (57.23) | 17895 (52.59) | 31176 (56.77) | 34518 (59.13) | 19680 (59.42) |
| Former smoker | 63446 (35.16) | 12138 (35.67) | 19337 (35.21) | 20268 (34.72) | 11703 (35.33) |
| Current smoker | 13267 (7.35) | 3896 (11.45) | 4255 (7.75) | 3450 (5.91) | 1666 (5.03) |
| Alcohol intake (%) | | | | | |
| Never | 10376 (5.75) | 1776 (5.22) | 2898 (5.28) | 3445 (5.90) | 2257 (6.81) |
| Special occasions only | 16955 (9.40) | 2919 (8.58) | 4876 (8.88) | 5504 (9.43) | 3656 (11.04) |
| One to three times a month | 19500 (10.81) | 3579 (10.52) | 5896 (10.74) | 6250 (10.71) | 3775 (11.40) |
| Once or twice a week | 45303 (25.11) | 8173 (24.02) | 13889 (25.29) | 14964 (25.63) | 8277 (24.99) |
| Three or four times a week | 46353 (25.69) | 8551 (25.13) | 14331 (26.09) | 15078 (25.83) | 8393 (25.34) |
| Daily or almost daily | 41828 (23.18) | 8998 (26.44) | 12987 (23.65) | 13093 (22.43) | 6750 (20.38) |
| Physical activity (%) | | | | | |
| None | 2216 (1.23) | 652 (1.92) | 686 (1.25) | 612 (1.05) | 266 (0.80) |
| Low | 24885 (13.79) | 5615 (16.50) | 8079 (14.71) | 7442 (12.75) | 3749 (11.32) |
| Moderate | 82185 (45.55) | 14833 (43.59) | 25035 (45.59) | 26920 (46.11) | 15397 (46.49) |
| High | 44204 (24.50) | 7780 (22.86) | 12918 (23.52) | 14613 (25.03) | 8893 (26.85) |
| Hypertension (%) | 76077 (42.16) | 15004 (44.09) | 23591 (42.96) | 24562 (42.07) | 12920 (39.01) |
| Body mass index (kg/m$^2$) | 26.82 ± 4.53 | 27.64 ± 4.65 | 27.08 ± 4.50 | 26.60 ± 4.46 | 25.95 ± 4.39 |
| Total energy intake (kcal/day) | 2046.9 ± 543.0 | 2191.3 ± 569.2 | 2095.9 ± 535.7 | 2021.2 ± 517.6 | 1862.2 ± 514.4 |
| Whole grains (g/day) | 130.0 (65.0, 162.0) | 130.0 (32.5, 130.0) | 130.0 (65.0, 162.5) | 130.0 (65.0, 173.3) | 130.0 (65.0, 173.3) |
| Potatoes (g/day) | 29.0 (0.0, 58.0) | 29.0 (0.0, 58.0) | 29.0 (0.0, 58.0) | 29.0 (0.0, 58.0) | 25.4 (0.0, 58.0) |
| Vegetables (g/day) | 200.0 (100.0, 350.0) | 100.0 (33.3, 175.0) | 158.3 (75.0, 348.0) | 258.3 (150.0, 400.0) | 318.8 (230.0, 462.5) |
| Fruits (g/day) | 244.8 (133.0, 381.0) | 75.0 (0.0, 210.0) | 218.3 (112.5, 339.0) | 300.0 (174.0, 417.0) | 324.0 (217.5, 472.5) |
| Dairy (g/day) | 49.0 (7.0, 154.7) | 28.0 (0.0, 118.0) | 42.0 (7.0, 142.0) | 60.0 (14.0, 177.0) | 74.0 (14.0, 180.0) |
| Red meat (g/day) | 40.0 (0.0, 80.0) | 65.0 (40.0, 80.0) | 50.0 (13.3, 80.0) | 40.0 (0.0, 80.0) | 0.0 (0.0, 26.7) |
| Poultry (g/day) | 0.0 (0.0, 40.0) | 0.0 (0.0, 60.0) | 0.0 (0.0, 40.0) | 0.0 (0.0, 40.0) | 0.0 (0.0, 26.7) |
| Eggs (g/day) | 0.0 (0.0, 30.0) | 30.0 (0.0, 60.0) | 0.0 (0.0, 30.0) | 0.0 (0.0, 15.0) | 0.0 (0.0, 0.0) |
| Fish (g/day) | 0.0 (0.0, 50.0) | 0.0 (0.0, 33.3) | 0.0 (0.0, 50.0) | 0.0 (0.0, 50.0) | 25.0 (0.0, 66.7) |
| Legumes (g/day) | 0.0 (0.0, 25.0) | 0.0 (0.0, 33.3) | 0.0 (0.0, 25.0) | 0.0 (0.0, 25.0) | 0.0 (0.0, 25.0) |
| Nuts (g/day) | 0.0 (0.0, 0.0) | 0.0 (0.0, 0.0) | 0.0 (0.0, 0.0) | 0.0 (0.0, 0.0) | 0.0 (0.0, 9.3) |
| Unsaturated fat: saturated fat | 1.5 (1.2, 1.9) | 1.5 (1.2, 1.8) | 1.5 (1.2, 1.8) | 1.5 (1.2, 1.9) | 1.7 (1.3, 2.1) |
| Added sugar (g/day) | 54.2 (35.8, 77.3) | 64.9 (46.4, 90.2) | 59.0 (41.8, 81.1) | 52.8 (35.5, 74.3) | 31.1 (21.6, 57.5) |

Data were presented as frequency (%), mean ± standard deviation or median (interquartile range).

that occurred within the first five years of follow-up (Supplementary Tables 9-11). Supplementary Tables 12–14 display the associations of individual components of the EAT-Lancet diet with risks of depression, anxiety and co-occurrence of depression and anxiety. In the Stubbendorff EAT-Lancet index, after full adjustment as in model 3 and additional adjustment for other individual food components, higher adherence to the vegetable recommendation was significantly associated with lower risks of depression, anxiety and their co-occurrence

(all $P$ for trend < 0.05). Similar associations were observed for the other two indexes. Mediation analyses showed that 22.10% (95% CI:14.60%–36.00%), 7.80% (95% CI: 3.29%–16.00%) and 18.50% (95% CI: 8.61%–60.00%) of the associations between the Knuppel EAT-Lancet index and depression, anxiety and their co-occurrence were mediated by BMI, respectively. The mediating effects of cardiovascular disease (CVD), type 2 diabetes (T2D) and hypertension were relatively weak or not significant. Similar results were observed for the Stubbendorff and

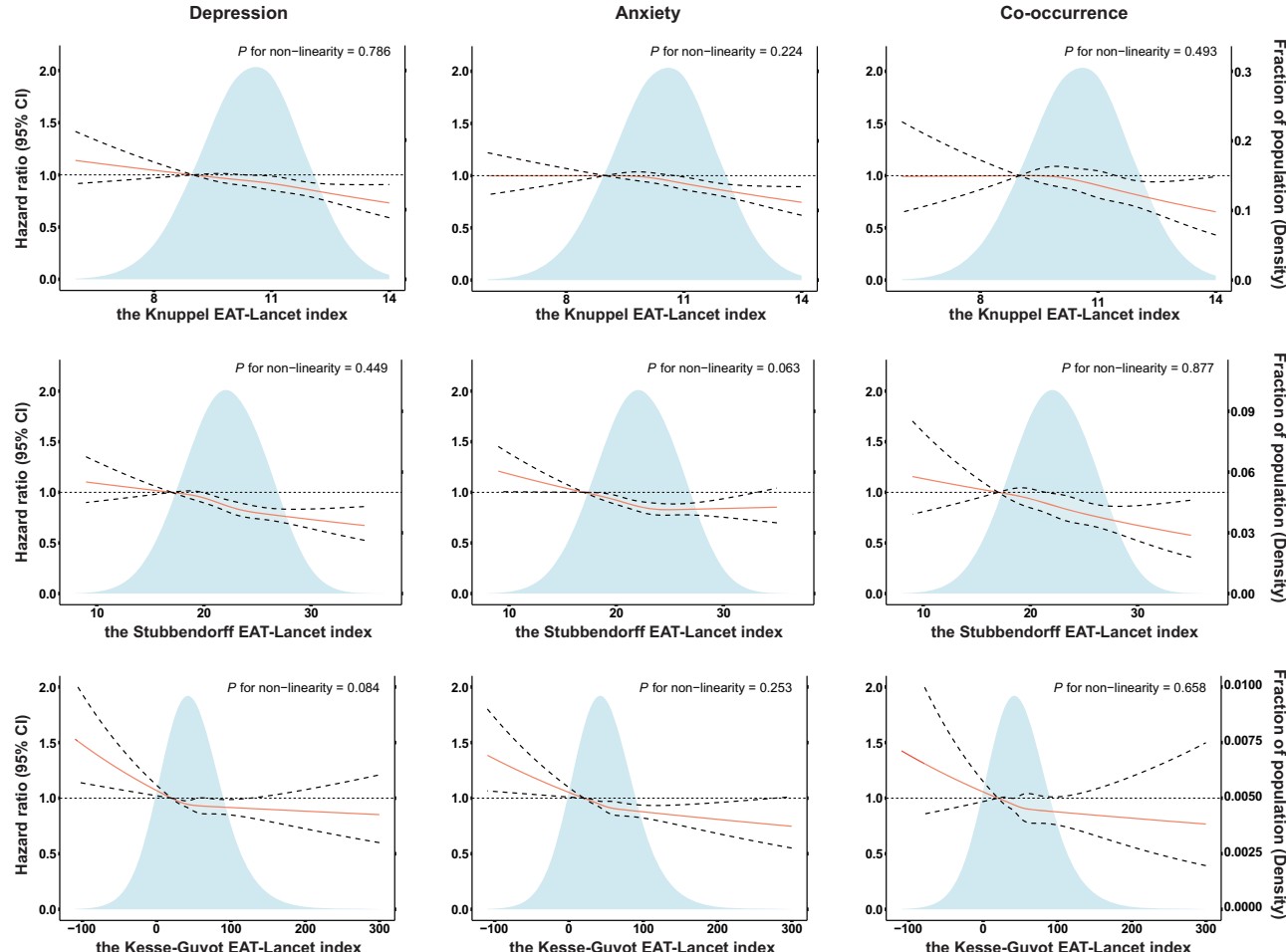

**Fig. 1 | The dose-response associations between the Knuppel, Stubbendorff and Kesse-Guyot EAT-Lancet index and risks of incident depression, anxiety and their co-occurrence.** Solid line: Point estimation; Dotted line: 95% confidence interval; Shading: Fraction of population (Density). Multivariable Cox regression models with restricted cubic splines were adjusted for age, sex, Townsend deprivation scores, ethnicity, smoking status, alcohol intake, physical activity, hypertension, total energy intake and BMI. References are 9 for the Knuppel EAT-Lancet index, 17 for the Stubbendorff EAT-Lancet index, and 20 (the cut-off value of quintile 1) for the Kesse-Guyot EAT-Lancet index. Statistical tests were two-sided and P-value of < 0.05 was considered statistically significant. No adjustments were made for multiple comparisons. Source data are provided as a Source Data file.

the Kesse-Guyot EAT-Lancet index (Supplementary Table 15). We also explored the potential interactions between the EAT-Lancet index and age, sex, smoking status and Townsend deprivation index. We found the associations of adhering to the EAT-Lancet index with incident depression and anxiety were more pronounced in those who were more deprived, and the results were consistent among the three EAT-Lancet indexes (all *P* for interaction <0.05) (Supplementary Tables 16–18).

## Discussion

In this prospective, population-based cohort with a median follow-up of 11.62 years, we found that greater adherence of the EAT-Lancet index was associated with lower risks of incident depression, anxiety and their co-occurrence. These associations remained consistent among the three different EAT-Lancet indexes.

The relationships between diet and depression and anxiety have gained considerable concern in recent years. Pooling all "healthy dietary patterns" from 17 studies, a 2018 meta-analysis demonstrated that adhering to a high-quality diet, regardless of type, was associated with a significantly reduced risk of depression (OR: 0.77, 95% CI: 0.69–0.84)[13]. More specifically, a large prospective study in UK Biobank showed that a plant-based diet rich in healthier plant foods was associated with a lower risk of incident depression (HR_Q5 vs. Q1: 0.92,

95% CI: 0.85–0.99) while a plant-based diet that emphasizes less-healthy plant foods was associated with a higher risk of incident depression (HR_Q5 vs. Q1: 1.15, 95% CI: 1.07–1.24)[18]. A cross-sectional study also suggested that a high-quality plant-based diet might be protective against depressive symptoms in Australian vegans and vegetarians[19]. Similar associations have been observed for anxiety[29]. In contrast, the prospective association of the Healthy Eating Index (HEI) with incident depressive symptoms was non-significant (HR: 0.96, 95% CI: 0.86–1.07)[30] and the associations of adherence to four previously established DASH diet indices with incident depression were inconsistent and dependent on the scoring system[31]. Our results indicated that adhering to the EAT-Lancet diet was significantly associated with lower risks of incident depression, anxiety and their co-occurrence, and the predictive performance of the EAT-Lancet index was modestly better than or comparable to other established diet scores. It is noteworthy that all the dietary scores were calculated only based on the UK Biobank, so the results were data-driven and future studies comparing the predictive performance of different diet scores in different settings and populations are needed.

Compared with the mainstream "healthy" diet patterns, the EAT-Lancet reference diet has also been designed to serve as an anchor for integrating sustainability into national dietary recommendations of culturally diverse countries[23]. Using data from the European

**Table 2 | Associations between the Knuppel EAT-Lancet index (range, 0-14) and risks of depression and anxiety**

| | N$_{case}$/N$_{total}$ | Model 0[a] | | Model 1[b] | | Model 2[c] | |
|---|---|---|---|---|---|---|---|
| | | HR (95% CI) | P-value | HR (95% CI) | P-value | HR (95% CI) | P-value |
| **Depression** | | | | | | | |
| EAT-Lancet diet index categories | | | | | | | |
| ≤ 9 | 968/34027 | REF | | REF | | REF | |
| = 10 | 1378/54919 | 0.879 (0.810–0.955) | 0.002 | 0.862 (0.793–0.936) | < 0.001 | 0.912 (0.839–0.990) | 0.029 |
| = 11 | 1445/58378 | 0.866 (0.798–0.939) | < 0.001 | 0.823 (0.757–0.893) | < 0.001 | 0.898 (0.826–0.976) | 0.011 |
| ≥ 12 | 757/33122 | 0.798 (0.725–0.877) | < 0.001 | 0.724 (0.657–0.798) | < 0.001 | 0.806 (0.730–0.890) | < 0.001 |
| P for trend | - | < 0.001 | | < 0.001 | | < 0.001 | |
| 1-point increment in diet score | 4548/180446 | 0.945 (0.921–0.969) | < 0.001 | 0.919 (0.896–0.943) | < 0.001 | 0.949 (0.925–0.974) | < 0.001 |
| **Anxiety** | | | | | | | |
| EAT-Lancet diet index categories | | | | | | | |
| ≤ 9 | 1175/34027 | REF | | REF | | REF | |
| = 10 | 1836/54919 | 0.965 (0.897–1.039) | 0.346 | 0.917 (0.852–0.987) | 0.021 | 0.946 (0.879–1.019) | 0.143 |
| = 11 | 1962/58378 | 0.969 (0.902–1.042) | 0.398 | 0.874 (0.812–0.941) | < 0.001 | 0.915 (0.850–0.985) | 0.019 |
| ≥ 12 | 1053/33122 | 0.916 (0.843–0.995) | 0.039 | 0.777 (0.714–0.846) | < 0.001 | 0.818 (0.751–0.892) | < 0.001 |
| P for trend | - | 0.066 | | < 0.001 | | < 0.001 | |
| 1-point increment in diet score | 6026/180446 | 0.984 (0.963–1.006) | 0.163 | 0.939 (0.918–0.960) | < 0.001 | 0.953 (0.932–0.975) | < 0.001 |
| **Co-occurrence** | | | | | | | |
| EAT-Lancet diet index categories | | | | | | | |
| ≤ 9 | 262/34027 | REF | | REF | | REF | |
| = 10 | 392/54919 | 0.924 (0.790–1.081) | 0.324 | 0.890 (0.760–1.041) | 0.145 | 0.940 (0.803–1.101) | 0.444 |
| = 11 | 406/58378 | 0.899 (0.770–1.050) | 0.179 | 0.828 (0.707–0.969) | 0.019 | 0.901 (0.769–1.057) | 0.201 |
| ≥ 12 | 202/33122 | 0.787 (0.655–0.946) | 0.011 | 0.685 (0.568–0.826) | < 0.001 | 0.756 (0.624–0.914) | 0.004 |
| P for trend | - | 0.013 | | < 0.001 | | 0.005 | |
| 1-point increment in diet score | 1262/180446 | 0.946 (0.902–0.993) | 0.024 | 0.909 (0.866–0.955) | < 0.001 | 0.937 (0.892–0.985) | 0.010 |

[a]Model 0: unadjusted.
[b]Model 1: adjusted for age, sex, Townsend scores and ethnicity.
[c]Model 2: adjusted for model 1 plus smoking status, alcohol intake, physical activity, hypertension, BMI and total energy intake.
Statistical tests were two-sided and P-value of < 0.05 was considered statistically significant. No adjustments were made for multiple comparisons.

Prospective Investigation into Cancer and Nutrition (EPIC) study, Laine et al. revealed that switching from lower adherence to the EAT-Lancet reference diet to higher adherence could potentially reduce food-associated greenhouse gas emissions up to 50% and land use up to 62%, and associations of levels of dietary greenhouse gas emissions with all-cause mortality (HR: 1.13, 95% CI: 1.10–1.16) and of land use with all-cause mortality (1.18, 1.15–1.21) were observed[32]. In addition, higher adherence to the EAT-Lancet reference diet was associated with lower risks of multiple diseases: type 2 diabetes[24,28], ischemic heart disease[26] and subarachnoid stroke[27], and all-cause[33,34] and cause-specific mortality[34]. Therefore, our current findings extended previous studies by uncovering inverse associations of adherence to the EAT-Lancet reference diet with depression, anxiety and their co-occurrence, supporting that the EAT-Lancet reference diet is not only favorable for the environment health and population's physical health, but also from a mental health aspect.

There were several approaches for translating the EAT-Lancet reference diet into the dietary scores. We adopted three common yet differently quantified EAT-Lancet indexes in the current study and the results were consistent, which indicates robust associations between adherence to the EAT-Lancet diet and incident mental outcomes. The differences among the three indices lie in the scoring criteria and food group categorizations. Compared with the binary assessments in the Knuppel EAT-Lancet index, the Kesse-Guyot EAT-Lancet index was a continuous score accounting for deviation from the cut-off value based on the Knuppel score and the Stubbendorff score used an ordinal scale, which discriminates different levels of adherence to the EAT-Lancet diet. As the emphasized individual components in all indexes, adherence to the vegetable and fruit recommendation was

significantly associated with lower risks of depression, anxiety and co-occurrence of depression and anxiety. Higher intake of fish, which is emphasized in the Stubbendorff index yet limited in other two indexes, was also observed to be inversely associated with all mental outcomes, which was supported by a previous study showing that n-3 polyunsaturated fatty acids contained in fish was inversely associated with depression[35]. Taken together, these discrepancies in score construction may explain the differences in the predictive performances among the three EAT-Lancet indexes.

Although depression and anxiety have been considered as two distinct entities according to the diagnostic criteria, co-occurrence of depression and anxiety is relatively a common syndrome[36]. Among the 4548 participants who developed depression in our study, 1262 (27.7%) participants also had anxiety. Previous reports have shown that compared with individuals with single depression or anxiety, those with their coexistence had a higher risk of suicidal ideation and previous suicide attempts and had poorer responses to treatment[37,38], indicating co-occurrence of depression and anxiety deserves particular concern on its management and treatment. Our study suggested that adhering to the EAT-Lancet reference diet may serve as a promising modifiable target which enhances the primary prevention for co-occurrence of depression and anxiety. In addition, we observed that the associations between adhering to the EAT-Lancet diet and incident depression and anxiety were more pronounced in those who were more deprived. Previous studies also suggested that a lower socioeconomic status was associated with greater risks of depression and anxiety[39,40]. Our findings suggest that the risk differences between different socioeconomic statuses might be narrowed by adhering to the EAT-Lancet diet.

**Table 3 | Associations between the Stubbendorff EAT-Lancet index (range, 0-38) and risks of depression and anxiety**

| | N_case/N_total | Model 0[a] | | Model 1[b] | | Model 2[c] | |
|---|---|---|---|---|---|---|---|
| | | HR (95% CI) | P-value | HR (95% CI) | P value | HR (95% CI) | P-value |
| **Depression** | | | | | | | |
| EAT-Lancet diet index categories | | | | | | | |
| ≤17 | 640/20312 | REF | | REF | | REF | |
| 18–20 | 1134/40578 | 0.883 (0.801–0.973) | 0.012 | 0.868 (0.787–0.956) | 0.004 | 0.926 (0.840–1.021) | 0.124 |
| 21–23 | 1348/55648 | 0.761 (0.693–0.836) | < 0.001 | 0.737 (0.670–0.810) | < 0.001 | 0.820 (0.745–0.903) | < 0.001 |
| 24–26 | 993/43019 | 0.724 (0.656–0.800) | < 0.001 | 0.685 (0.619–0.758) | < 0.001 | 0.788 (0.712–0.873) | < 0.001 |
| ≥ 27 | 433/20889 | 0.648 (0.574–0.732) | < 0.001 | 0.601 (0.531–0.681) | < 0.001 | 0.711 (0.627–0.806) | < 0.001 |
| *P* for trend | - | < 0.001 | | < 0.001 | | < 0.001 | |
| 1-point increment in diet score | 4548/180446 | 0.967 (0.959–0.975) | < 0.001 | 0.961 (0.953–0.969) | < 0.001 | 0.974 (0.966–0.982) | < 0.001 |
| **Anxiety** | | | | | | | |
| EAT-Lancet diet index categories | | | | | | | |
| ≤17 | 770/20312 | REF | | REF | | REF | |
| 18-20 | 1408/40578 | 0.912 (0.835–0.996) | 0.040 | 0.868 (0.794–0.948) | 0.002 | 0.902 (0.826–0.986) | 0.023 |
| 21-23 | 1819/55648 | 0.856 (0.787–0.931) | < 0.001 | 0.788 (0.724–0.858) | < 0.001 | 0.838 (0.769–0.913) | < 0.001 |
| 24-26 | 1380/43019 | 0.839 (0.768–0.916) | < 0.001 | 0.745 (0.681–0.815) | < 0.001 | 0.806 (0.736–0.883) | < 0.001 |
| ≥27 | 649/20889 | 0.810 (0.730–0.899) | < 0.001 | 0.701 (0.631–0.779) | < 0.001 | 0.765 (0.687–0.852) | < 0.001 |
| *P* for trend | - | < 0.001 | | < 0.001 | | < 0.001 | |
| 1-point increment in diet score | 6026/180446 | 0.986 (0.979–0.993) | < 0.001 | 0.974 (0.968–0.981) | < 0.001 | 0.981 (0.974–0.988) | < 0.001 |
| **Co-occurrence** | | | | | | | |
| EAT-Lancet diet index categories | | | | | | | |
| ≤17 | 172/20312 | REF | | REF | | REF | |
| 18-20 | 322/40578 | 0.934 (0.776–1.124) | 0.469 | 0.901 (0.748–1.085) | 0.271 | 0.962 (0.798–1.159) | 0.683 |
| 21-23 | 365/55648 | 0.768 (0.641–0.921) | 0.004 | 0.722 (0.602–0.868) | < 0.001 | 0.804 (0.668–0.968) | 0.021 |
| 24-26 | 290/43019 | 0.788 (0.653–0.952) | 0.014 | 0.718 (0.593–0.870) | < 0.001 | 0.826 (0.680–1.003) | 0.054 |
| ≥27 | 113/20889 | 0.631 (0.498–0.800) | <0.001 | 0.561 (0.441–0.713) | < 0.001 | 0.659 (0.516–0.841) | < 0.001 |
| *P* for trend | - | < 0.001 | | < 0.001 | | < 0.001 | |
| 1-point increment in diet score | 1262/180446 | 0.967 (0.953–0.982) | < 0.001 | 0.958 (0.944–0.973) | < 0.001 | 0.971 (0.956–0.986) | < 0.001 |

[a]Model0: unadjusted.
[b]Model1: adjusted for age, sex, Townsend scores and ethnicity.
[c]Model2: adjusted for model 1 plus smoking status, alcohol intake, physical activity, hypertension, BMI and total energy intake.
Statistical tests were two-sided and *P*-value of < 0.05 was considered statistically significant. No adjustments were made for multiple comparisons.

The underlying mechanisms for the associations of the EAT-Lancet index with depression and anxiety have not been fully elucidated but may be explained from several aspects. First, this mainly plant-based diet pattern similar with the Mediterranean dietary pattern may reduce markers of inflammation in humans[41]. The inflammatory effects of a diet high in calories and saturated fat have been proposed as one mechanism that has detrimental effects on brain health, including cognitive decline, hippocampal dysfunction, and damage to the blood-brain barrier[42,43]. Various mental health condition, have been also linked to heightened inflammation[44]. Second, the way that food may affect mental wellbeing has been recently explained by the effect of dietary patterns on the gut microbiome. The gut microbiome interacts with the brain in bidirectional ways by neural, inflammatory, and hormonal signaling pathways[45]. Consumption of a diet high in fibers, polyphenols, and unsaturated fatty acids can promote gut microbial taxa which can metabolize these food sources into anti-inflammatory metabolites[43,46] that regulate emotion in the human brain. At last, future studies investigating the extent to which the EAT-Lancet diet indexes are correlated or interplay with the Mediterranean Diet score, the DASH diet score, the HEI, the inflammatory diet score and other plant-based diet scores will definitely help to provide a more in-depth insight into the potential mechanism.

In this study, we examined the associations among three different versions of the EAT-Lancet diet index and risks of depression and anxiety, verifying the robustness of the associations. There are also five occasions of baseline assessments throughout a year which enables us to account for seasonal differences of food components intake and calculate an average number as a marker of habitual intake. Other strengths of this study include the large sample size and a prospective study design. Our study also has several potential limitations. First, although multiple sources of medical conditions were used in the current study to identify the occurrence of depression and anxiety, a proportion of participants may still go underdiagnosed or not be diagnosed in time[47], and the presence of symptoms may alter the dietary intakes. However, individual dietary habit is usually maintained lifelong and less likely to change greatly over time[48]. Second, more than half of the population was excluded because they have not completed the 24-hour dietary recall questionnaire on any occasion. Although the characteristics between the analysis population and the total population were statistically different (Supplementary Table 19), these differences were minor from a clinical perspective, and we have adjusted all these potential influencing factors in multivariate analyses. Third, diet is a time-varying exposure and the ability of our study to detect these changes was limited. However, measurement error is unlikely to ever be entirely eliminated from dietary assessment[49–51] and using multiple 24 h dietary recall surveys has utmost reduced the measurement error. Forth, the 42-points Stubbendorff score did not fit the 24-hour dietary recall in UK Biobank perfectly because there was limited information about the unsaturated oils and red meat when

**Table 4 | Associations between the Kesse-Guyot EAT-Lancet index and risks of depression and anxiety**

| | N_case/N_total | Model 0[a] | | Model 1[b] | | Model 2[c] | |
|---|---|---|---|---|---|---|---|
| | | HR (95% CI) | *P*-value | HR (95% CI) | *P* value | HR (95% CI) | *P*-value |
| Depression | | | | | | | |
| EAT-Lancet diet index categories | | | | | | | |
| Quintile 1 | 1050/36090 | REF | | REF | | REF | |
| Quintile 2 | 874/36088 | 0.826 (0.755–0.903) | < 0.001 | 0.810 (0.740–0.886) | < 0.001 | 0.874 (0.799–0.957) | 0.004 |
| Quintile 3 | 855/36089 | 0.807 (0.738–0.884) | < 0.001 | 0.775 (0.708–0.849) | < 0.001 | 0.859 (0.784–0.941) | 0.001 |
| Quintile 4 | 874/36090 | 0.824 (0.754–0.902) | < 0.001 | 0.769 (0.702–0.843) | < 0.001 | 0.862 (0.786–0.945) | 0.002 |
| Quintile 5 | 895/36089 | 0.846 (0.774–0.925) | < 0.001 | 0.760 (0.693–0.833) | < 0.001 | 0.844 (0.768–0.928) | < 0.001 |
| *P* for trend | - | 0.001 | | < 0.001 | | 0.001 | |
| 100-point increment in diet score | 4548/180446 | 0.872 (0.805–0.944) | < 0.001 | 0.790 (0.728–0.858) | < 0.001 | 0.864 (0.795–0.938) | < 0.001 |
| Anxiety | | | | | | | |
| EAT-Lancet diet index categories | | | | | | | |
| Quintile 1 | 1288/36090 | REF | | REF | | REF | |
| Quintile 2 | 1180/36088 | 0.910 (0.841–0.984) | 0.019 | 0.869 (0.803–0.941) | < 0.001 | 0.911 (0.842–0.987) | 0.022 |
| Quintile 3 | 1152/36089 | 0.888 (0.820–0.961) | 0.003 | 0.816 (0.753–0.884) | < 0.001 | 0.867 (0.800–0.940) | < 0.001 |
| Quintile 4 | 1191/36090 | 0.918 (0.848–0.993) | 0.033 | 0.809 (0.747–0.877) | < 0.001 | 0.864 (0.797–0.937) | < 0.001 |
| Quintile 5 | 1215/36089 | 0.938 (0.867–1.014) | 0.107 | 0.784 (0.724–0.850) | < 0.001 | 0.825 (0.759–0.896) | < 0.001 |
| *P* for trend | - | 0.171 | | < 0.001 | | < 0.001 | |
| 100-point increment in diet score | 6026/180446 | 0.957 (0.893–1.025) | 0.206 | 0.811 (0.756–0.871) | < 0.001 | 0.842 (0.784–0.905) | < 0.001 |
| Co-occurrence | | | | | | | |
| EAT-Lancet diet index categories | | | | | | | |
| Quintile 1 | 281/36090 | REF | | REF | | REF | |
| Quintile 2 | 256/36088 | 0.904 (0.763–1.071) | 0.242 | 0.874 (0.737–1.035) | 0.119 | 0.948 (0.799–1.124) | 0.537 |
| Quintile 3 | 230/36089 | 0.812 (0.682–0.966) | 0.019 | 0.760 (0.638–0.906) | 0.002 | 0.846 (0.709–1.009) | 0.064 |
| Quintile 4 | 252/36090 | 0.888 (0.749–1.053) | 0.172 | 0.802 (0.675–0.953) | 0.012 | 0.900 (0.756–1.072) | 0.238 |
| Quintile 5 | 243/36089 | 0.859 (0.723–1.020) | 0.083 | 0.739 (0.620–0.882) | < 0.001 | 0.818 (0.682–0.981) | 0.030 |
| *P* for trend | - | 0.095 | | 0.001 | | 0.030 | |
| 100-point increment in diet score | 1262/180446 | 0.878 (0.754–1.021) | 0.092 | 0.765 (0.654–0.894) | < 0.001 | 0.832 (0.711–0.973) | 0.021 |

[a]Model 0: unadjusted.
[b]Model 1: adjusted for age, sex, Townsend scores and ethnicity.
[c]Model 2: adjusted for model 1 plus smoking status, alcohol intake, physical activity, hypertension, BMI and total energy intake.
Statistical tests were two-sided and *P*-value of < 0.05 was considered statistically significant. No adjustments were made for multiple comparisons.

**Table 5 | Net reclassification improvement for the risks of depression and anxiety associated with different diet indexes**

| | Depression | | Anxiety | | Co-occurrence | |
|---|---|---|---|---|---|---|
| | Estimate (95% CI) | *P*-value | Estimate (95% CI) | *P*-value | Estimate (95% CI) | *P*-value |
| Reference model + Knuppel EAT-Lancet index | 0.038 (0.008–0.064) | < 0.001 | 0.012 (−0.010 to 0.044) | 0.277 | 0.038 (−0.008 to 0.088) | 0.139 |
| Reference model + Stubbendorff EAT-Lancet index | 0.112 (0.082–0.138) | < 0.001 | 0.088 (0.062–0.116) | < 0.001 | 0.146 (0.058–0.202) | < 0.001 |
| Reference model + Kesse-Guyot EAT-Lancet index | 0.060 (0.026–0.092) | < 0.001 | 0.044 (0.018–0.072) | < 0.001 | 0.090 (0.010–0.136) | < 0.001 |
| Reference model + PDI | 0.020 (−0.014 to 0.052) | 0.139 | 0.000 (−0.018 to 0.030) | 0.911 | 0.044 (−0.004 to 0.106) | 0.079 |
| Reference model + hPDI | 0.040 (−0.002 to 0.070) | 0.059 | 0.004 (−0.018 to 0.034) | 0.614 | 0.052 (−0.018 to 0.106) | 0.099 |
| Reference model + uPDI | 0.112 (0.080–0.142) | < 0.001 | 0.084 (0.060–0.106) | < 0.001 | 0.142 (0.088–0.190) | < 0.001 |
| Reference model + DASH index | 0.082 (0.052–0.112) | < 0.001 | 0.026 (0.000–0.052) | 0.059 | 0.078 (0.012–0.132) | 0.020 |
| Reference model + MD index | 0.042 (0.012–0.060) | < 0.001 | 0.024 (−0.002 to 0.054) | 0.079 | 0.030 (−0.020 to 0.088) | 0.436 |

Reference model includes age, sex, Townsend scores, ethnicity, smoking status, alcohol intake, physical activity, hypertension, total energy intake and BMI.
*PDI* Overall plant-based diet index, *hPDI* healthful plant-based diet index, *uPDI* Unhealthful plant-based diet index, *DASH* Dietary Approaches to Stop Hypertension, *MD* Mediterranean diet.
Statistical tests were two-sided and *P*-value of < 0.05 was considered statistically significant. No adjustments were made for multiple comparisons.

processed meat intake was assessed. Nonetheless, separating red meat into beef/lamb and pork was mainly for environmental reasons, as beef/lamb has approximately five times greater greenhouse gas emission than pork[52], which was less connected with our main topic. Despite this 4-points discrepancy, it did not substantially affect the ability of our 38-points index to distinguish participants with different levels of adherence to the EAT-Lancet reference diet. Last, this population-based cohort from the UK Biobank might not fully represent the general population. Since over 95% of participants in our study are White, we repeated the main analysis in White and observed similar

results (Supplementary Table 20). Whether the present findings could be generalized to other populations need to be further verified.

In conclusion, this prospective study revealed that higher adherence to the EAT-Lancet reference diet, reflected by three EAT-Lancet diet scores, was associated with lower risks of depression, anxiety and their co-occurrence. The findings highlight that promoting this achievable and sustainable dietary pattern might have far-reaching benefits for the prevention of depression and anxiety.

## Methods

### Study design and population
The UK Biobank study was approved by the North West Multi-Centre Research Ethics Committee (REC reference: 11/NW/03820). Written informed consent were provided by all participants. The UK Biobank is a large and prospective study with over 500,000 participants aged 37–73 years (99.5% aged 40–69 years) recruited from the general population between 2006-2010[53]. After providing written informed consent, participants completed a touch screen questionnaire and a face-to-face interview and underwent a series of physical measurements and biological sample collection at any one of the 22 assessment centers across England, Scotland and Wales[53].

For the current study, participants who had completed the online 24 h dietary recall questionnaire on at least one occasion were eligible for inclusion. Participants who met any of the following criteria were excluded: (1) withdrew from the survey; (2) suffered from depression or anxiety at baseline; (3) reported use of anxiolytics or antidepressants at baseline; (4) with abnormal total energy intakes ( < 500 or > 3500 kcal/day in female and < 800 or > 4000 kcal/day in male participants)[24]. Finally, a total of 180,446 participants were eligible for subsequent analyses (Supplementary Fig. 1).

### Dietary assessment and EAT-Lancet diet index
Dietary information was collected via a dietary questionnaire called the Oxford WebQ, based on a 24 h dietary recall of the previous day (https://biobank.ndph.ox.ac.uk/showcase/refer.cgi?id=118240), which is similar to a 24-hour dietary recall assessing the types and quantities of foods consumed, including beverages and daily nutrient intake, which has been validated in details elsewhere[49,54]. The questionnaire was first introduced as part of the assessment visit towards the end of recruitment for the last 70,000 participants (2009-2010), and participants were invited to complete 4 additional questionnaires online at 3-4 monthly intervals on four separate occasions between Feb 2011 and April 2012. If participants completed dietary assessments multiple times, we used all the dietary intake data to calculate an average intake to account for variations in dietary intake.

The EAT-Lancet diet index was calculated to estimate the adherence to the EAT-Lancet Commission recommendations on healthy diets from sustainable food systems[22]. There are several approaches for translating the EAT-Lancet diet into the dietary scores and there is currently no consensus of how to quantify adherence to the EAT-Lancet reference diet. To assess the robustness of the relationship between adhering to the EAT-Lancet diet and incident depression, anxiety and co-occurrence of depression and anxiety, we adopted three different diet scores in our current study. The Knuppel score was the most widely used score for health outcomes[26], and the Stubbendorff score was the only score using an ordinal scale to score each component[25], which allows for a more thorough measurement of the degrees of adherence to each dietary component and the overall EAT-Lancet diet. The Kesse-Guyot score was a continuous score based on the component and cut-off defined by Knuppel et al., which is able to capture interindividual variability[55]. The score by Knuppel et al. contains 8 main dietary categories, including whole grains, tubers and starchy vegetables, vegetables, fruits, dairy foods, protein sources, added fats, and added sugars. Participants were assigned with 1 or 0 point for meeting or failing each of the recommendations, which

resulted in a total score of 0-14 points. The score by Stubbendorff et al. is consisted of 14 dietary components, with 7 emphasized components (whole grains, vegetables, fruits, fish, legumes, nuts, and unsaturated oils) and 7 limited components (potatoes, dairy, beef and lamb, pork, poultry, eggs, and added sugar). Details of this EAT-Lancet diet index were described in the development article[25]. Participants in this study were assigned with 3 points for the highest adherence and 0 point for the lowest for emphasized components and the scoring pattern was inverted for the limited components, which constructed the EAT-Lancet index ranging from 0 (worst) to 38 (best) points, with higher scores indicating greater adherence to the recommended eating pattern. In the score by Kesse-Guyot et al., the value considered the deviation from the cut-off and the calculation of this score leads to a continuous variable (no set range, positive or negative). The higher the score, the more in line the individual's diet is with the EAT-Lancet recommendations. Details of this dietary score have been previously described[55]. To distinguish the three scoring criteria in this study, we defined the EAT-Lancet diet index by Knuppel et al. as the Knuppel EAT-Lancet diet index, the EAT-Lancet diet index by Stubbendorff et al. as the Stubbendorff EAT-Lancet diet index and the EAT-Lancet diet index by Kesse-Guyot et al. as the Kesse-Guyot EAT-Lancet diet index. Detailed assessments of the three EAT-Lancet indexes and other dietary patterns can be found in Supplementary Methods 1. Definition of portion size and food items used in this study can be seen in Supplementary Table 21. The cut-off for each component of the Knuppel and the Kesse-Guyot EAT-Lancet diet index can be seen in Supplementary Table 22 and the cut-off of the Stubbendorff EAT-Lancet diet index can be seen in Supplementary Table 23. Supplementary Tables 24 and 25 display the numbers of participants in each category of individual components of the Knuppel and Stubbendorff EAT-Lancet diet indexes. Supplementary Table 26 displays the median (IQR) of scores of each individual components of the Kesse-Guyot EAT-Lancet diet index.

### Outcome identification
Disease outcomes were identified from self-reported medical conditions, primary care, linked inpatient hospital data, and death registry records from "first occurrence fields" linked to the UK Biobank (https://biobank.ndph.ox.ac.uk/showcase/refer.cgi?id=593). Follow-up occurrences of depression and anxiety were ascertained by the International Statistical Classification of Diseases and Related Health Problems, Tenth Revision (ICD-10) codes F32-F33 and F40-F48, respectively[5,56]. The co-occurrence of depression and anxiety was defined as developing both depression and anxiety during follow-up. The follow-up period for participants was from the date of completion of the baseline assessment to the date of occurrence of the outcome, date of death, or the last date of follow-up (March 23, 2021), whichever came first.

### Assessment of covariates
Covariates included sociodemographic characteristics, lifestyle factors and other potential confounding factors and were collected at baseline, except for total energy intake which was assessed by the 24-hour dietary recall questionnaire. Detailed description of the covariates can be found in Supplementary Methods 2.

### Statistical analysis
For baseline characteristics, we used mean ± SD or median (IQR) to describe continuous variables and frequencies (percentages) to describe categorical variables. Kaplan-Meier curves stratified by the EAT-Lancet index are shown in Supplementary Fig. 2. Cox proportional hazards models were used to estimate HRs and 95% CIs for the associations between the baseline EAT-Lancet index and new-onset depression, anxiety and co-occurrence of depression and anxiety. The proportional hazards assumption was checked using Schoenfeld

residuals. Dose-response relationships were examined by restricted cubic splines with four nodes[57].

To ensure adequate numbers of participants in each group, the participants were divided into four groups for the Knuppel score ($\leq 9$, $= 10$, $= 11$, $\geq 12$ points) and five groups for the Stubbendorff score ($\leq 17$, 18–20, 21–23, 24–26, and $\geq 27$ points), respectively. Participants were divided into quintiles according to the Kesse-Guyot score. With the lowest group as the reference, we ran a total of three models: model 0 was unadjusted; model 1 was adjusted for age, sex, Townsend scores and ethnicity; and model 2 was adjusted for model 1 plus smoking status, alcohol intake, physical activity, hypertension, BMI and total energy intake.

To compare the predictive performance of the EAT-Lancet index and other established diet scores (the Mediterranean diet score, the DASH diet score and the plant-based diet scores) for incident mental disorders, NRI index was calculated. The NRI index could assess the proportion of cases that would be reclassified correctly by adding each diet score to the reference model[58]. We set the truncation time at the median follow-up time.

We performed a series of sensitivity and subgroup analyses to evaluate the robustness of our primary findings. First, we excluded those who completed the online 24 h dietary recall questionnaire on only one occasion to obtain the average intake of all dietary components as a marker of habitual intake. Second, since the four additional online dietary assessment took place later than the baseline assessment, we repeated the main analysis using the follow-up time which began at the time when the latest dietary assessment was completed. Third, considering the potential impact of reverse causality, we excluded depression, anxiety or co-occurrence cases that occurred within the first five years of follow-up. Fourth, to further explore the different roles of individual components in the EAT-Lancet index, we tested the associations between individual components and the outcomes. Fifth, mediation analyses were performed to investigate the possible pathways between adherence to EAT-Lancet diet and risks of mental disorders, and we calculated the mediation proportions (95% CIs) of BMI, CVD, T2D and hypertension. Sixth, potential interaction effects were tested by adding interaction terms between the EAT-Lancet diet indexes and covariates (age, sex, socioeconomic status and smoking status) in the model and stratified subgroup analyses were also performed.

Statistical analyses were performed using the SAS version 9.4 (SAS Institute) and R (version 4.1.1 software). All statistical tests were two-sided, and $P$-value of $< 0.05$ was considered statistically significant.

### Reporting summary
Further information on research design is available in the Nature Portfolio Reporting Summary linked to this article.

## Data availability
Data are available in a public, open access repository. This research has been conducted using the UK Biobank Resource under Application Number 60651. The data that support the findings of this study are available on application to the UK Biobank team at http://www. ukbiobank.ac.uk/. Source data are provided with this paper.

## Code availability
The simulation codes of SAS version 9.4 and R version 4.1.1 for statistical analysis of this study are available at https://github.com/Luxujia/ UKB_EAT-Lancet_and_Depression_and_Anxiety.

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

## Acknowledgements

This work was supported by the National Natural Science Foundation of China (81703316) and A Project Funded by Priority Academic Program Development of Jiangsu Higher Education Institutions (PAPD). The authors would like to thank all participants and staff from the UK Biobank cohort.

## Author contributions

The authors' responsibilities were as follows - C.K., Y.B.: were responsible for the study's concept and design; X.L., L.W.: performed statistical analysis; X.L., C.K.: drafted the manuscript; X.L., L.S., Y.F.: interpreted the results of statistical analysis; C.K., Y.B., X.L., Y.P., X.L.: revised the manuscript critically for important intellectual contents; and all authors: agreed to be accountable for all aspects of the work and read and approved the final manuscript.

## Competing interests

The authors declare no competing interests.
