## [Peer Review File · Nature Communications]

Adherence to the EAT-Lancet Diet and Incident Depression and AnxietyReviewers' Comments:

Reviewer #2:

Remarks to the Author:

This is an interesting manuscript about an important topic. However, the need for this study should be better described, as well as some methodological considerations and limitations. It should also be very clear throughout the manuscript (and maybe even in the title) that the outcome is clinical diagnosis of depression/anxiety, as these are commonly underdiagnosed conditions.

1. Introduction: It should focus more on the need for this study. Why consider a sustainable diet? Other studies have evaluated other dietary patterns and their association with depression with mixed results. What food groups are new in this pattern?

2. Methods: The length of follow-up is not detailed in the methods. The year when the repeated 24hr diet recalls is considered baseline, but the timeline for the outcome assessment is not clear. Later in the discussion it is mentioned that it is 11.6 y. But it is not known how long the gap was between dietary assessments and depression/anxiety diagnosis. The authors reported a sensitivity analysis with a 2 year gap between dietary assessment and diagnosis. I would suggest to use this as the main results and add another sensitivity analysis with a 5 year gap between dietary assessment and diagnosis. The problem is that diagnosis symptoms can occur for a long time before diagnosis.

3. Discussion: The important limitation of using clinical diagnosis of depression/anxiety is not discussed. As mentioned above, many cases go underdiagnosed and the presence of symptoms can alter the dietary intakes.

Reviewer #3:

Remarks to the Author:

Révision

In this article entitled "Adherence to the EAT-Lancet Diet and Incident Depression and Anxiety: A Prospective Cohort Study", authors examined the association between adherence to EAT-Lancet reference diet proposed by the EAT-Lancet Commission in 2019 and risk of depression, anxiety and co-morbidity.

Literature based on prospective observational cohort framework evidenced robust association between adherence to several dietary recommendations (including Mediterranean diet, Alternative healthy Eating Index, DASH scores) and onset, severity and recurrence of depressive and anxiety disorders. Based on recent development of nutritional epidemiology research, authors propose to apply new dietary recommendations proposed by the EAT-Lancet commission into the field of common mental disorders.

To do so, they used data from the large UK biobank study (500 000 participants, aged 37-73 years) recruited from general population. Carrying research to identify the most efficient dietary guidelines to reduce the risk of depression and anxiety disorders remains crucial in the new field of nutritional psychiatry research, thus the scope of the current paper is relevant and raises interest. In the following review, I would like to provide suggestions and comments to authors. Hoping this might help to improve clarity and scientific impact of the manuscript.

INTRODUCTION

In the first paragraph, authors focused on epidemiology of common mental disorders - depression and anxiety- and its impact on DALYs, mortality and morbidity. I would suggest authors to mention how the Covid pandemic has made common mental disorders' public health issues eminently crucial.

L.57 : "diet has been recommended as promising modifiable targets for the prevention and treatment of mental disorders[9][10]." authors should replace recommended by suggested. Authors should cite here meta-analyses and principle papers investigated association between overall diet and common mental health disorders (CMD) instead of the reference 10. Please add a sentence explaining why diet can be an interesting candidate as a public health target in a context

of low efficiency of available treatments and etiology of CMD.

L 59 - 62 : "when examining the association between diet and mental disorders" authors added two references not related to mental health- ref 11 and 12.

L61 - 64 : please correct the reference citations It seems that 11 and 12 ones should be cited I 64 instead of the 13 -14 references.

I think the second section of the introduction can be improved. Overall the scientific rationale of the present study is not enough clearly stated. As exposed, readers can have the impression that authors applied the EAT Lancet diet indices to mental health without any hypothesis. Authors stated the EAT Lancet diet \diamond CVD risk / T2D, EAT Lancet diet \diamond physical activity, and the increased risk of CVD with mental disorders (actually it is a bidirectional association). What are the authors' hypothesis regarding EAT Lancet diet and common mental disorders association? (For instance it could be Eat lancet \diamond Physical activity \diamond common mental disorders (CMD) or CMD as mediator of the Eat Lancet- CVD, or CVD mediator of the EAT Lancet- CMD, or common causes ?)

If authors have no pathway hypothesis, then the aim is to assess the extent to which the EAT-Lancet dietary pattern predicts new onset of common mental disorders and whether this diet is a better predictor compared to other established dietary scores.

Additional suggestion in this introduction section: authors should describe and explain the concept of the EAT lancet diet.

L71 : Here authors should refer to the literature on plant based diet and depression (Wu H, Gu Y, Meng G, et al. Quality of plant-based diet and the risk of dementia and depression among middle-aged and older population. *Age Ageing* 2023;52(5):afad070 ; Plant-based dietary quality and depressive symptoms in Australian vegans and vegetarians: a cross-sectional study M. Lee, Ryan Eather and T. Best)

L73. " In this prospective cohort study, we aimed to investigate the associations between adherence to the EAT-Lancet reference diet and new-onset clinically confirmed diagnoses of depression and anxiety in a large population-based cohort of UK adults." In this paper association with comorbidity has also been investigated I suggest to add this in the objective statement. In addition, for more clarity I would recommend the use of co-occurrence instead of co-morbidity

Minor typo error : Introduction, l.52 correct "severer".

METHODS

Study Design

Data came from the impressive cohort of UK biobank where baseline took place in 2006-2010. However, I would be grateful to authors to provide some additional details regarding the design of their analyses.

I understood that participants with depression or anxiety disorders at baseline (2006-2010) have been excluded.

"Participants who completed the dietary questionnaire on at least one occasion were eligible for inclusion". Could authors list the years when these dietary questionnaires were proposed? How did authors proceed when several questionnaires were available? Line 159 (stat section) I understand that diet was assessed several times and average intakes have been computed. It would be helpful to provide more details.

I think additional material to help readers understand the follow-up phases would be welcome too. How did authors make sure that diet questionnaires were filled before the assessment of depression and anxiety disorders? I saw the sensitivity analyses in which incident cases over the two first years were excluded? But this did not completely resolve the reverse causation question.

Section on dietary intakes and EAT Lancet diet

Minor comment: ref 19 is mentioned twice. In addition when I consulted ref 19, the link to access to the diet questionnaire did not work. I encourage the authors to add an updated link in the current paper.

L 90 . Would it be possible to put these values in kcal/day, it makes comparisons with other studies easier.

L 102 - 107 : Why did authors not use the EAT-Lancet Diet Index (ELD-I) which offers a continuous scoring for each component. The EFD-I has been shown to better capture inter-individual variability and thus has greater discriminant power than scores only considering discrete or even binary scoring for each component. Please refer to <https://doi.org/10.1093/ajcn/nqac208>.

I would like to see results when computing this index.

Outcome identification

How has medication (anxiolytics and antidepressants) been considered?

How has comorbidity been defined? Is it having depression and anxiety at the same time? Or having depression or anxiety. Please provide details.

How much cases have been defined as self-reported, registry linkage and death registry records?

Covariates assessments

Authors should specify when covariates have been collected? ie: At the same phase of dietary questionnaire?

Could authors explain why dietary habits were included as covariates? I do not understand the rationale of having these variables as covariates.

Statistical analysis & RESULTS

L 145-147: "Cox proportional hazards models were used to estimate hazard ratios (HRs) and 95% confidence intervals (CIs) for the associations between the baseline EAT-Lancet index and new-onset depression, anxiety and their comorbidity."

"Comorbidity": this has not been defined in methods. Please clarify, see my previous comment

L 156: total energy intakes have not been mentioned in covariates section.

L 161 Regarding sensitivity analyses on directionality and reverse causation, please see my previous comment.

L 161: Before carrying subgroup analyses as sensitivity analyses, could authors provide test of interactions?

Some additional sensitivity analyses should be added and tested:

The mediating role of BMI, CVD, in the EAT-Lancet and common mental disorders incidence

The potential interaction effect of smoking, ethnicity and diet with regards to CMD

The extent to which EAT Lancet diet predicts better than other plant-based diet already proposed in literature

The assessment of each component of the EAT Lancet Diet –CMD association

Regarding the co-occurrence, could we have estimates for onset of depression alone (after excluding anxiety) and anxiety alone (after excluding depression) and for onset of any CMD (anxiety or depression)?

Results

L 175: please add indications on co-occurrence of depression and anxiety.

Analyses of associations between individual components and CMD would be relevant.

Above I suggested few additional analyses, I encourage authors to display the results.

DISCUSSION

The discussion regarding the novelty and validation of EAT-Lancet diet should be enriched.

I think authors should specify how the EAT –Lancet has been built (or specify this in Intro), on which bases, its targets. i.e. While the EAT-Lancet diet provides a set of recommendations for feeding the entire world population within planetary boundaries, it has been designed to serve as an anchor for integrating sustainability into national dietary recommendations of culturally diverse countries.

We expect some discussion regarding components, the cut-offs used, which ones seem adapted to mental health outcomes and why? some additional comparison with similar plant-based diet and depression association previously published is lacking. Please refer to the ref I mentioned above (p.2)

In addition, line 278, authors mentioned publications assessing EAT lancet diet and chronic diseases. Authors mentioned studies reporting significant associations, what about those failing to observe any association between EAT Lancet diet and mortality (Knuppel et al .). What do authors think about this following paper? Zgmutt FJ, Pouzou JG, Costard S The EAT-Lancet Commission's dietary composition may not prevent non communicable disease mortality J Nutr., 150 (5) (2020), pp. 985-988.

Authors dedicated a section considering potential mechanisms. Authors mentioned inflammation

and gut microbiome as stated in all papers reporting research in nutritional psychiatry. By deriving other diet indices, authors would be able to provide how the EAT Lancet diet scores are correlated with Med Diet, DASH, AHEI, Inflammatory diet scores. At least this should be put as perspective. Given the richness of the cohort, we would expect more discussion regarding who benefit the most of adhering to EAT Lancet diet in regards to CMD.

What about the mediating role of BMI, CVD, hypertension or T2D? Does adherence in EAT lancet diet different according to ethnic group all these elements would definitely contribute to improve the current paper.

Thank you very much for your comments and professional advice, all of which have helped us to significantly improve the quality of our manuscript.

Reviewer #2 (Remarks to the Author):

Comment #1: This is an interesting manuscript about an important topic. However, the need for this study should be better described, as well as some methodological considerations and limitations. It should also be very clear throughout the manuscript (and maybe even in the title) that the outcome is clinical diagnosis of depression/anxiety, as these are commonly underdiagnosed conditions.

Response: Thank you for appreciating our work and offering constructive comments. Incident depression and anxiety were identified from International Statistical Classification of Diseases and Related Health Problems, Tenth Revision (ICD-10) codes in the current study. Since depression and anxiety are commonly underdiagnosed conditions, we used multiple sources of medical conditions to identify the occurrence of depression and anxiety (linked inpatient hospital data, primary care, self-reported medical conditions, and death registry records, which were integrated as the “first occurrence fields (Category ID: 1712)” in the UK Biobank). We used the word “clinical” in our original manuscript to distinguish the depression and anxiety cases defined in our study from symptom-based depression or anxiety. However, after carefully reviewing literatures in the related field, we found “clinical” was rarely used unless the cases were only defined by hospital admission records[1]. The identification in our study by self-reported (although participants were asked whether they had been

diagnosed (by a doctor) with each condition they self-reported), primary care and death registry records may not all meet the strict clinical diagnosis standards. So, after careful consideration, we decided not to use the word “clinical” and defined the cases in our study as “incident depression and anxiety” from “first occurrence fields” in the UK Biobank, which have also been previously used [2][3], and we also described their definitions in more details in the revised manuscript on Pages 19-20, Lines 380-387. We agree with you that depression and anxiety may be underdiagnosed in this study. We carefully addressed the problem of the potential underdiagnosis in Response to Comment 4 below.

Comment #2: Introduction: It should focus more on the need for this study. Why consider a sustainable diet? Other studies have evaluated other dietary patterns and their association with depression with mixed results. What food groups are new in this pattern?

Response: Thank you for your constructive comments. Sustainable diets aim to increase both human and planetary health. As a sustainable diet, the EAT-Lancet reference diet takes into account the multiple links between health, nutrition intake and the environment, and provides a set of recommendations for feeding the global population within planetary boundaries[4]. A large European cohort study demonstrated that switching from lower adherence to the EAT–Lancet reference diet to higher adherence could potentially reduce food-associated greenhouse gas emissions up to 50% and land use up to 62%[5]. Another recent study indicated that the EAT-Lancet

reference diet was associated with less freshwater eutrophication, less marine eutrophication, less terrestrial acidification, and higher blue water use[6]. This dietary pattern emphasizes the intake of vegetables, fruits, whole grains, and nuts, and limits the intake of animal source foods, added sugar, and saturated fats[7]. Compared with Mediterranean Diet, the EAT-Lancet diet focuses more on legumes and cereals[8] and there are also important differences with other diet patterns due to the different designs of the scores, such as intake ranges and cut-off values used. We have emphasized the need of our study in our revised manuscript, which can be found on Pages 4-5, Lines 52-69.

Comment #3: Methods: The length of follow-up is not detailed in the methods. The year when the repeated 24hr diet recalls is considered baseline, but the timeline for the outcome assessment is not clear. Later it in the discussion it is mentioned that it is 11.6 y. But it is not known how long the gap was between dietary assessments and depression/anxiety diagnosis. The authors reported a sensitivity analysis with a 2 year gap between dietary assessment and diagnosis. I would suggest to use this as the main results and add another sensitivity analysis with a 5 year gap between dietary assessment and diagnosis. The problem is that diagnosis symptoms can occur for a long time before diagnosis.

Response: Thank you for your comments. The follow-up in our study began at the time when baseline assessment was completed and we censored participants at the date of occurrence of the outcome, date of death, or the last date of follow-up (March 23, 2021),

whichever came first, which has been mentioned in the outcome identification section (Page 20, Lines 387-389).

The dates of occurrence of outcomes in the UK Biobank (UKB) were available through the linkage of all UKB participants to their health-related records including self-reported medical conditions, primary care, inpatient hospital data, and death registry records. Details of these health-related records can be found on the UK Biobank website (<https://biobank.ndph.ox.ac.uk/showcase/refer.cgi?id=593>).

We did not use the results of the “2 year gap” sensitivity analysis as the main results so that the current results could be more comparable with most existing researches[3][9][10]. Based on your advice, we have performed another sensitivity analysis where we excluded depression, anxiety or co-occurrence cases which occurred within the first five years of follow-up, and the results were virtually unchanged. In the revised manuscript, we have replaced the “2 year gap” sensitivity analysis with the “5 year gap” sensitivity analysis, which can be found on Pages 8-9, Lines 152-153 and Pages 21-22, Lines 424-426.

Comment #4: Discussion: The important limitation of using clinical diagnosis of depression/anxiety is not discussed. As mentioned above, many cases go underdiagnosed and the presence of symptoms can alter the dietary intakes.

Response: Thank you very much for your suggestion. We agree with you that the ICD-based depression and anxiety instead of symptom-based depression and anxiety may result in underestimation, and the presence of symptoms may alter the dietary intakes. However, studies have showed that individual dietary habit is usually maintained

lifelong and less likely to change greatly over time[11].

Moreover, we performed additional analysis where we randomly selected 10% of the individuals with incident depression, anxiety or the co-occurrence of depression and anxiety (case group), respectively, and recategorized them into the control group to simulate the underdiagnosed conditions. The results were not materially changed and are presented in the **Table 1** below.

In our revised manuscript, we have added this as one important limitation on Pages 14-15, Lines 274-282.

Table 1 Associations between the EAT-Lancet indexes and risks of depression and anxiety after randomly recategorizing 10% of individuals in the case group into the control group

	Depression	Anxiety	Co-occurrence
	HR (95% CI)	HR (95% CI)	HR (95% CI)
Knuppel EAT-Lancet index			
≤9	REF	REF	REF
=10	0.914 (0.837-0.997)	0.948 (0.877-1.025)	0.941 (0.797-1.111)
=11	0.902 (0.826-0.985)	0.917 (0.848-0.992)	0.904 (0.764-1.069)
≥12	0.810 (0.730-0.899)	0.820 (0.749-0.899)	0.757 (0.619-0.926)
P for trend	0.0002	<0.0001	0.0061
1-point increment in diet score	0.949 (0.923-0.975)	0.953 (0.930-0.976)	0.938 (0.890-0.989)
Stubbendorff EAT-Lancet index			
≤17	REF	REF	REF
18-20	0.927 (0.836-1.027)	0.905 (0.824-0.993)	0.975 (0.800-1.187)
21-23	0.821 (0.742-0.909)	0.841 (0.768-0.921)	0.823 (0.677-1.000)
24-26	0.789 (0.709-0.879)	0.810 (0.736-0.891)	0.852 (0.693-1.046)
≥27	0.712 (0.623-0.812)	0.769 (0.687-0.861)	0.679 (0.524-0.878)
P for trend	<0.0001	<0.0001	0.0012
1-point increment in diet score	0.974 (0.965-0.982)	0.981 (0.974-0.989)	0.974 (0.958-0.990)
Kesse-Guyot EAT-Lancet index			
Quintile 1	REF	REF	REF
Quintile 2	0.875 (0.795-0.962)	0.915 (0.842-0.995)	0.951 (0.794-1.139)
Quintile 3	0.861 (0.782-0.948)	0.872 (0.801-0.949)	0.851 (0.706-1.026)
Quintile 4	0.866 (0.785-0.954)	0.870 (0.799-0.948)	0.903 (0.751-1.086)
Quintile 5	0.849 (0.769-0.938)	0.832 (0.762-0.908)	0.823 (0.679-0.997)
P for trend	0.0025	<0.0001	0.0444
100-point increment in diet score	0.866 (0.794-0.944)	0.847 (0.785-0.913)	0.840 (0.712-0.991)

Models were adjusted for age, sex, Townsend scores, ethnicity, smoking status, alcohol intake, physical activity, hypertension, total energy intake and BMI.

References

- [1] Dregan A, Rayner L, Davis KAS, et al. Associations Between Depression, Arterial Stiffness, and Metabolic Syndrome Among Adults in the UK Biobank Population Study: A Mediation Analysis. *JAMA Psychiatry*. 2020;77(6):598-606.
- [2] Yang T, Wang J, Huang J, Kelly FJ, Li G. Long-term Exposure to Multiple Ambient Air Pollutants and Association With Incident Depression and Anxiety. *JAMA Psychiatry*. 2023;80(4):305-313.
- [3] Sun M, He Q, Li G, et al. Association of ultra-processed food consumption with incident depression and anxiety: a population-based cohort study. *Food Funct*. 2023;14(16):7631-7641.
- [4] Berthy F, Brunin J, Allès B, et al. Association between adherence to the EAT-Lancet diet and risk of cancer and cardiovascular outcomes in the prospective NutriNet-Santé cohort. *Am J Clin Nutr*. 2022;116(4):980-991.
- [5] Laine JE, Huybrechts I, Gunter MJ, et al. Co-benefits from sustainable dietary shifts for population and environmental health: an assessment from a large European cohort study. *Lancet Planet Health*. 2021;5(11):e786-e796.
- [6] Colizzi C, Harbers MC, Vellinga RE, et al. Adherence to the EAT-Lancet Healthy Reference Diet in Relation to Risk of Cardiovascular Events and Environmental Impact: Results From the EPIC-NL Cohort. *J Am Heart Assoc*. 2023;12(8):e026318.
- [7] Willett W, Rockström J, Loken B, et al. Food in the Anthropocene: the EAT-Lancet Commission on healthy diets from sustainable food systems. *Lancet*.

2019;393(10170):447-492.

- [8] Carcel C, Bushnell C. Can Dietary Patterns That Support Planetary Health Benefit Population Health?. *Stroke*. 2022;53(1):164-166.
- [9] Xu C, Cao Z, Huang X, Wang X. Associations of healthy lifestyle with depression and post-depression dementia: A prospective cohort study. *J Affect Disord*. 2023;327:87-92.
- [10] Kaiser A, Schaefer SM, Behrendt I, Eichner G, Fasshauer M. Association of sugar intake from different sources with incident depression in the prospective cohort of UK Biobank participants. *Eur J Nutr*. 2023;62(2):727-738.
- [11] Goode AD, Reeves MM, Eakin EG. Telephone-delivered interventions for physical activity and dietary behavior change: an updated systematic review. *Am J Prev Med*. 2012;42(1):81-88.

Reviewer #3 (Remarks to the Author):

In this article entitled “Adherence to the EAT-Lancet Diet and Incident Depression and Anxiety: A Prospective Cohort Study”, authors examined the association between adherence to EAT-Lancet reference diet proposed by the EAT-Lancet Commission in 2019 and risk of depression, anxiety and co-morbidity.

Literature based on prospective observational cohort framework evidenced robust association between adherence to several dietary recommendations (including Mediterranean diet, Alternative healthy Eating Index, DASH scores) and onset, severity and recurrence of depressive and anxiety disorders. Based on recent development of nutritional epidemiology research, authors propose to apply new dietary recommendations proposed by the EAT-Lancet commission into the field of common mental disorders.

To do so, they used data from the large UK biobank study (500 000 participants, aged 37-73 years) recruited from general population. Carrying research to identify the most efficient dietary guidelines to reduce the risk of depression and anxiety disorders remains crucial in the new field of nutritional psychiatry research, thus the scope of the current paper is relevant and raises interest.

In the following review, I would like to provide suggestions and comments to authors. Hoping this might help to improve clarity and scientific impact of the manuscript.

Response: Thank you for appreciating our work and offering constructive comments.

The answers to your valuable comments are provided below:

INTRODUCTION

Comment #1: In the first paragraph, authors focused on epidemiology of common mental disorders - depression and anxiety- and its impact on DALYs, mortality and morbidity. I would suggest authors to mention how the Covid pandemic has made common mental disorders' public health issues eminently crucial.

Response: Thank you very much for your suggestion. Based on your kind recommendations, we have mentioned how the COVID-19 pandemic has made the public health issues of depression and anxiety eminently crucial[1][2] in the first paragraph, which can be found on Page 3, Lines 36-39.

Comment #2: L.57: “diet has been recommended as promising modifiable targets for the prevention and treatment of mental disorders[9][10].” authors should replace recommended by suggested. Authors should cite here meta-analyses and princeps papers investigated association between overall diet and common mental health disorders (CMD) instead of the reference 10. Please add a sentence explaining why diet can be an interesting candidate as a public health target in a context of low efficiency of available treatments and etiology of CMD.

Response: Thank you for your suggestion. In the revised manuscript, we have replaced the word “recommended” with “suggested” and replaced the reference 10 by a systematic review and meta-analysis[3]. We have also explained why diet can be an interesting candidate as a public health target in a context of low efficiency of available treatments and etiology of CMD[4][5], which can be found on Pages 3-4, Lines 43-49.

Comment #3: L 59 - 62: “when examining the association between diet and mental disorders” authors added two references not related to mental health– ref 11 and 12.

Response: Thank you for pointing this out. We have replaced the references 11 and 12 with two articles investigating the association between diet and mental disorders[6][7].

Comment #4: L61 - 64: please correct the reference citations It seems that 11 and 12 ones should be cited l 64 instead of the 13 -14 references.

Response: Thank you for your comment. We have corrected the references 11 and 12 based on your kind advice above. The original references 13 and 14, which were about the relationships between the EAT-Lancet diet and health outcomes, were supposed to be cited there.

Comment #5: I think the second section of the introduction can be improved. Overall the scientific rationale of the present study is not enough clearly stated. As exposed, readers can have the impression that authors applied the EAT Lancet diet indices to mental health without any hypothesis. Authors stated the EAT Lancet diet - CVD risk / T2D, EAT Lancet diet - physical activity, and the increased risk of CVD with mental disorders (actually it is a bidirectional association). What are the authors’ hypothesis regarding EAT Lancet diet and common mental disorders association? (For instance it could be Eat lancet - Physical activity - common mental disorders (CMD) or CMD as mediator of the Eat Lancet- CVD, or CVD mediator of the EAT Lancet - CMD, or common causes?)

If authors have no pathway hypothesis, then the aim is to assess the extent to which the EAT-Lancet dietary pattern predicts new onset of common mental disorders and whether this diet is a better predictor compared to other established dietary scores.

Additional suggestion in this introduction section: authors should describe and explain the concept of the EAT lancet diet.

Response: Thank you for your constructive comments. Based on your advice, we have added the associations of other established dietary scores (i.e. Mediterranean diet score[8], DASH diet score[9] and plant-based diet score[10][11]) with depression and/or anxiety. We did not intend to have any pathway hypothesis, so we aimed to evaluate the extent to which adhering to the EAT-Lancet diet was associated with risks of incident depression and anxiety. In addition, we have described and explained the concept of the EAT lancet diet in this introduction section. Details can be found in our revised manuscript (Pages 4-5, Lines 52-72).

Comment #6: L71: Here authors should refer to the literature on plant based diet and depression (Wu H, Gu Y, Meng G, et al. Quality of plant-based diet and the risk of dementia and depression among middle-aged and older population. *Age Ageing* 2023;52(5):afad070 ; Plant-based dietary quality and depressive symptoms in Australian vegans and vegetarians: a cross-sectional study M. Lee, Ryan Eather and T. Best)

Response: Thank you for your comment. We have referred to the two articles in the revised manuscript on Page 4, Lines 52-55.

Comment #7: L73. “In this prospective cohort study, we aimed to investigate the associations between adherence to the EAT-Lancet reference diet and new-onset clinically confirmed diagnoses of depression and anxiety in a large population-based cohort of UK adults.” In this paper association with comorbidity has also been investigated I suggest to add this in the objective statement. In addition, for more clarity I would recommend the use of co-occurrence instead of co-morbidity.

Response: Thank you for your kind suggestion. We have replaced the word “co-morbidity” with “co-occurrence” throughout the revised manuscript and added this in the objective statement (Page 5, Lines 69-72).

Comment #8: Minor typo error: Introduction, l.52 correct “severer”.

Response: Thank you for pointing out this typo error. We have corrected it in the revised manuscript on Page 3, Line 36.

METHODS

Comment #9: Data came from the impressive cohort of UK biobank where baseline took place in 2006-2010. However, I would be grateful to authors to provide some additional details regarding the design of their analyses. I understood that participants with depression or anxiety disorders at baseline (2006-2010) have been excluded.

“Participants who completed the dietary questionnaire on at least one occasion were eligible for inclusion”. Could authors list the years when these dietary questionnaires were proposed? How did authors proceed when several questionnaires were available?

Line 159 (stat section) I understand that diet was assessed several times and average intakes have been computed. It would be helpful to provide more details.

Response: Thank you very much for your comments. The 24h dietary recall questionnaire was first introduced as part of the baseline assessment visit towards the end of recruitment for the last 70,000 participants (2009-2010). Participants who had provided UK Biobank with e-mail addresses were also invited, via e-mail, to complete the questionnaire online on four separate occasions between Feb 2011 and April 2012:

1st e-mail invitations: Feb 2011 - April 2011

2nd email invitations: June 2011 - Aug 2011

3rd email invitations: Oct 2011 – Dec 2011

4th email invitations: April 2012 – June 2012

If participants completed dietary assessments multiple times, the study used all the dietary intake data to calculate an average intake to account for variations in dietary intake. For example, if a participant completed 3 dietary recalls in total and reported ate 1, 2 and 1 servings of vegetable in the 3 recalls, then an average number of servings of vegetable per day was 1.33. In the revised manuscript, we have added more detailed descriptions found on Page 17, Lines 331-336.

The covariates in the UK Biobank were only assessed during baseline assessment (2006-2010), so the follow-up time in our study began at the time when baseline assessment was conducted. As a result, a small fraction of depression/anxiety cases might be identified before the later rounds of dietary assessment. To resolve this problem, we performed a sensitivity analysis using the follow-up phases which began

at the time when the latest dietary assessment was completed. In addition, we performed another sensitivity analysis in which we excluded cases that occurred within the first five years of follow-up and the results were virtually unchanged. Details can be seen in Response to Comment #11 below.

Comment #10: I think additional material to help readers understand the follow-up phases would be welcome too.

Response: Thank you for your comment. The follow-up phases in our study began at the time when baseline assessment was conducted and we censored participants at the date of occurrence of the outcome, date of death, or the last date of follow-up (March 23, 2021), whichever came first.

The dates of occurrence of outcomes in the UK Biobank (UKB) were available through the linkage of all UKB participants to their health-related records including self-reported medical conditions, primary care, inpatient hospital data, and death registry records. Details of these health-related records can be found on the UK Biobank website (<https://biobank.ndph.ox.ac.uk/showcase/refer.cgi?id=593>). In the revised manuscript, relevant information has been described on Pages 19-20, Lines 380-389.

Comment #11: How did authors make sure that diet questionnaires were filled before the assessment of depression and anxiety disorders? I saw the sensitivity analyses in which incident cases over the two first years were excluded? But this did not completely resolve the reverse causation question.

Response: Thank you for your questions. Considering that the covariates in the UK Biobank were only assessed at baseline (2006-2010), the follow-up in our study began from the date of the baseline assessment. However, the 4 additional online dietary assessment (Feb 2011 and April 2012) took place later, which led to that a small fraction of cases might be identified before the later rounds of dietary assessment. To resolve this problem, we performed a sensitivity analysis using the follow-up phases which began at the time when the latest dietary assessment was completed. Additionally, to minimize the possibility of reverse causation, we added another sensitivity analysis where we excluded cases which occurred within the first five years of follow-up. The results of the two sensitivity analyses remained consistent with the main results, and the associations persisted in both sensitivity analyses (Pages 8-9, Lines 150-153).

Comment #12: Minor comment: ref 19 is mentioned twice. In addition when I consulted ref 19, the link to access to the diet questionnaire did not work. I encourage the authors to add an updated link in the current paper.

Response: Thank you for pointing out this issue. We have corrected it in the revised manuscript and added a link of the 24h dietary recall questionnaire in UK Biobank in the revised manuscript (<https://biobank.ndph.ox.ac.uk/showcase/refer.cgi?id=118240>).

Comment #13: L 90. Would it be possible to put these values in kcal/day, it makes comparisons with other studies easier.

Response: Thank you for your suggestion. We have put the values of total energy intake

in kcal/day in our revised manuscript.

Comment #14: L 102 - 107: Why did authors not use the EAT-Lancet Diet Index (ELD-I) which offers a continuous scoring for each component. The ELD-I has been shown to better capture inter-individual variability and thus has greater discriminant power than scores only considering discrete or even binary scoring for each component. Please refer to <https://doi.org/10.1093/ajcn/nqac208>.

I would like to see results when computing this index.

Response: Thank you so much for your constructive comments. We have calculated the ELD-I and found significant associations of ELD-I with incident depression, anxiety and co-occurrence of depression and anxiety. Compared with the lowest quintile of ELD-I, the highest quintile was associated with lower risks of depression, anxiety and co-occurrence of depression and anxiety (all *P* trend <0.05), with respective HRs (95% CIs) of 0.844 (0.768-0.928), 0.825 (0.759-0.896) and 0.818 (0.682-0.981) (**Table 1**). We have added all relevant contents of ELD-I into our revised manuscript as one of the main results of our study.

Table 1 Associations between the ELD-I and risks of depression, anxiety and co-occurrence of depression and anxiety

	N _{case} /N _{total}	HR (95% CI)		
		Model 0 ¹	Model 1 ²	Model 2 ³
Depression				
EAT-Lancet diet index categories				
Quintile 1	1050/36090	REF	REF	REF
Quintile 2	874/36088	0.826 (0.755-0.903)	0.810 (0.740-0.886)	0.874 (0.799-0.957)
Quintile 3	855/36089	0.807 (0.738-0.884)	0.775 (0.708-0.849)	0.859 (0.784-0.941)
Quintile 4	874/36090	0.824 (0.754-0.902)	0.769 (0.702-0.843)	0.862 (0.786-0.945)
Quintile 5	895/36089	0.846 (0.774-0.925)	0.760 (0.693-0.833)	0.844 (0.768-0.928)

P for trend	-	0.0006	<0.0001	0.0009
100-point increment in diet score	4548/180446	0.872 (0.805-0.944)	0.790 (0.728-0.858)	0.864 (0.795-0.938)
Anxiety				
EAT-Lancet diet index categories				
Quintile 1	1288/36090	REF	REF	REF
Quintile 2	1180/36088	0.910 (0.841-0.984)	0.869 (0.803-0.941)	0.911 (0.842-0.987)
Quintile 3	1152/36089	0.888 (0.820-0.961)	0.816 (0.753-0.884)	0.867 (0.800-0.940)
Quintile 4	1191/36090	0.918 (0.848-0.993)	0.809 (0.747-0.877)	0.864 (0.797-0.937)
Quintile 5	1215/36089	0.938 (0.867-1.014)	0.784 (0.724-0.850)	0.825 (0.759-0.896)
P for trend	-	0.1707	<0.0001	<0.0001
100-point increment in diet score	6026/180446	0.957 (0.893-1.025)	0.811 (0.756-0.871)	0.842 (0.784-0.905)
Co-occurrence				
EAT-Lancet diet index categories				
Quintile 1	281/36090	REF	REF	REF
Quintile 2	256/36088	0.904 (0.763-1.071)	0.874 (0.737-1.035)	0.948 (0.799-1.124)
Quintile 3	230/36089	0.812 (0.682-0.966)	0.760 (0.638-0.906)	0.846 (0.709-1.009)
Quintile 4	252/36090	0.888 (0.749-1.053)	0.802 (0.675-0.953)	0.900 (0.756-1.072)
Quintile 5	243/36089	0.859 (0.723-1.020)	0.739 (0.620-0.882)	0.818 (0.682-0.981)
P for trend	-	0.0950	0.0006	0.0295
100-point increment in diet score	1262/180446	0.878 (0.754-1.021)	0.765 (0.654-0.894)	0.832 (0.711-0.973)

¹Model0: unadjusted.

²Model1: adjusted for age, sex, Townsend scores and ethnicity.

³Model2: adjusted for model 1 plus smoking status, alcohol intake, physical activity, hypertension, BMI and total energy intake.

Outcome identification

Comment #15: How has medication (anxiolytics and antidepressants) been considered?

Response: Thank you for your comment. We did not take medication use into account when assessing baseline depression and anxiety in our original manuscript. Based on your comment, we have further excluded individuals who reported taking anxiolytics or antidepressants at baseline and the subsequent associations were unchanged, and we have used this as the main results in our revised manuscript.

Comment #16: How has comorbidity been defined? Is it having depression and anxiety

at the same time? Or having depression or anxiety. Please provide details.

Response: Thank you for your questions. In our study, individuals with co-occurrence of depression and anxiety were defined as developing both depression and anxiety during follow-up, and we have added descriptions in the outcome identification section explaining it (Page 20, Lines 385-387).

Comment #17: How much cases have been defined as self-reported, registry linkage and death registry records?

Response: Thank you for this question. The number of cases defined as self-reported medical conditions, primary care, linked inpatient hospital data, and death registry records are presented in **Table 2** below.

Table 2 Source of report of depression and anxiety in UK Biobank

	Depression	Anxiety
Source of report		
Hospital admissions data only	3266 (72.22)	3626 (60.59)
Primary care only	660 (14.60)	1927 (32.20)
Self-report only	338 (7.47)	113 (1.89)
Primary care and other source	114 (2.52)	129 (2.16)
Self-report and other source	109 (2.41)	160 (2.67)
Hospital admissions data and other source	32 (0.71)	29 (0.48)
Death register only	3 (0.07)	0 (0.00)
Death register and other source	0 (0.00)	0 (0.00)

Covariates assessments

Comment #18: Authors should specify when covariates have been collected? ie: A the same phase of dietary questionnaire?

Response: Thank you for your comment. Covariates were collected only during

baseline assessment (except for total energy intake), which was at the same time of the first occasion of dietary questionnaire. Total energy intake was also assessed by the 24-hour dietary recall questionnaire. We have added this information in our revised manuscript on Page 20, Lines 391-394.

Comment #19: Could authors explain why dietary habits were included as covariates?

I do not understand the rationale of having these variables as covariates.

Response: Thank you for pointing out this issue. We did not adjust for dietary habits in our analysis, and we have corrected it in the revised manuscript accordingly.

STATISTICAL ANALYSIS & RESULTS

Comment #20: L 145-147: “Cox proportional hazards models were used to estimate hazard ratios (HRs) and 95% confidence intervals (CIs) for the associations between the baseline EAT-Lancet index and new-onset depression, anxiety and their comorbidity.”

“Comorbidity”: this has not been defined in methods. Please clarify, see my previous comment

Response: Thank you for your comment. The co-occurrence of depression and anxiety was defined as developing both depression and anxiety during follow-up, which has been clarified in the revised outcome identification section (Page 20, Lines 385-387).

Comment #21: L 156: total energy intakes have not been mentioned in covariates

section.

Response: Thank you for pointing out this issue. We have added total energy intake in the revised covariates section.

Comment #22: L 161 Regarding sensitivity analyses on directionality and reverse causation, please see my previous comment.

Response: Based on your previous comment, we have performed a sensitivity analysis calculating the follow-up time which began at the time when the latest dietary assessment was completed. Additionally, to minimize the possibility of reverse causation, we added another sensitivity analysis excluding cases within the first five years of follow-up. The results of the two sensitivity analyses remained consistent with the main results (Pages 8-9, Lines 150-153).

Comment #23: L 161: Before carrying subgroup analyses as sensitivity analyses, could authors provide test of interactions?

Response: Thank you for your comment. We have added an interaction term of each covariate with the EAT-Lancet index in the model to test the potential interaction effects, which can be found on Page 22, Lines 431-433. We found the associations of adhering to the EAT-Lancet index with incident depression and anxiety were more pronounced in those who were more deprived, and the results were consistent among the three EAT-Lancet indexes (all P for interaction <0.05), which can be found on Page 9, Lines 166-170.

Some additional sensitivity analyses should be added and tested:

Comment #24: The mediating role of BMI, CVD, in the EAT-Lancet and common mental disorders incidence.

Response: Based on your comment, we have tested the mediating roles of BMI, CVD, T2D and hypertension in the associations between the EAT-Lancet diet and depression and anxiety. Results are presented in **Table 3** below. We observed that 22.10% (95% CI:14.60%-36.00%), 7.80% (95% CI: 3.29%-16.00%) and 18.50% (95% CI: 8.61%-60.00%) of the associations between the Knuppel EAT-Lancet index and depression, anxiety and their co-occurrence were mediated by BMI, respectively. The mediating effects of CVD, T2D and hypertension were relatively weak or not significant. Similar results were observed for the Stubbendorff and the Kesse-Guyot EAT-Lancet index. Details can be found on Page 9, Lines 159-166.

Table 3 Mediating effects of BMI, CVD, T2D and hypertension in the associations between the EAT-Lancet indexes and risks of depression and anxiety

	Mediation proportion (95% CI), %			
	BMI	CVD	T2D	Hypertension
Knuppel EAT-Lancet index				
Depression	22.10 (14.60-36.00)	1.88 (0.90-4.00)	-1.18 (-2.71- 0.00)	-0.50 (-1.71-0.00)
Anxiety	7.80 (3.29-16.00)	1.57 (0.76-3.00)	0.17 (-0.68-1.00)	0.45 (0.00-1.00)
Co-occurrence	18.50 (8.61-60.00)	1.40 (0.49-5.00)	-0.08 (-1.95-1.00)	-0.63 (-5.32-0.00)
Stubbendorff EAT-Lancet index				
Depression	17.80 (13.00-27.00)	1.24 (0.68-2.00)	-0.55(-1.17-0.00)	-0.38 (-0.99-0.00)
Anxiety	7.50 (3.58-15.00)	1.32 (0.61-2.00)	0.01 (-0.47-1.00)	0.38 (-0.01-1.00)
Co-occurrence	14.10 (7.89-31.00)	0.98 (0.38-3.00)	-0.01 (-0.91-1.00)	-0.63 (-1.85-0.00)
Kesse-Guyot EAT-Lancet index				
Depression	23.1 (14.3-41.00)	1.79 (0.74-4.00)	-1.50 (-3.71 - -1.00)	-0.52 (-1.67-0.00)
Anxiety	6.12 (2.44-13.00)	1.20 (0.53-3.00)	0.18 (-0.62-1.00)	0.36 (0.00-1.00)
Co-occurrence	17.4 (7.19-76.00)	1.31 (0.39-6.00)	-0.12 (-2.36-3.00)	-0.06 (-4.57-0.00)

Models were adjusted for age, sex, Townsend scores, ethnicity, smoking status, alcohol intake, physical activity, total energy intake, and BMI, CVD, T2D and hypertension when these were not

considered the potential mediator.

Comment #25: The potential interaction effect of smoking, ethnicity and diet with regards to CMD

Response: We agree with you that the potential interaction effect of smoking and ethnicity needs to be tested. We have added an interaction term of smoking group with the EAT-Lancet index in the model to test the potential interaction effects. No significant interactions were found between EAT-Lancet diet indexes and smoking status groups (P for interaction >0.05), which can be found on Page 9, Lines 166-170. However, over 95% of participants in our study are White, so we repeated the main analysis in White and observed similar results, which have been added to Supplementary materials of the revised manuscript. We also discussed the generalizability of our current findings, which can be found on Pages 15-16, Lines 298-302.

Comment #26: The extent to which EAT Lancet diet predicts better than other plant-based diet already proposed in literature.

Response: Based on your comment, we have calculated the Mediterranean diet (MD) index, the DASH diet index and three plant-based diet quality indexes (an overall plant-based diet index, a healthy plant-based diet index, and an unhealthy plant-based diet index). We have also calculated the net reclassification improvement index (NRI) to measure and compare the predictive abilities of the EAT-Lancet diet and other established diet patterns. **Table 4** shows the NRI values of different diet patterns, and

the Stubbendorff and the Kesse-Guyot EAT-Lancet index had better predictive performance among the three EAT-Lancet indexes and were modestly better than or comparable to other established dietary scores. However, the findings need to be interpreted with caution since all the dietary scores were calculated only based on the UK Biobank, so the results were data-driven and future studies comparing the predictive performance of different diet scores in different settings and populations are needed. Details can be found on Pages 7-8, Lines 129-145.

Table 4 Net reclassification improvement for the risks of depression, anxiety and their co-occurrence associated with different diet scores

	Depression		Anxiety		Co-occurrence	
	Estimate (95% CI)	P	Estimate (95% CI)	P	Estimate (95% CI)	P
Reference model + Knuppel EAT-Lancet index	0.038 (0.008-0.064)	<0.001	0.012 (-0.010-0.044)	0.277	0.038 (-0.008-0.088)	0.139
Reference model + Stubbendorff EAT-Lancet index	0.112 (0.082-0.138)	<0.001	0.088 (0.062-0.116)	<0.001	0.146 (0.058-0.202)	<0.001
Reference model + Kesse-Guyot EAT-Lancet index	0.060 (0.026-0.092)	<0.001	0.044 (0.018-0.072)	<0.001	0.090 (0.010-0.136)	<0.001
Reference model + PDI	0.020 (-0.014-0.052)	0.139	0.000 (-0.018-0.030)	0.911	0.044 (-0.004-0.106)	0.079
Reference model + hPDI	0.040(-0.002-0.070)	0.059	0.004 (-0.018-0.034)	0.614	0.052 (-0.018-0.106)	0.099
Reference model + uPDI	0.112 (0.080-0.142)	<0.001	0.084 (0.060-0.106)	<0.001	0.142 (0.088-0.190)	<0.001
Reference model + DASH index	0.082 (0.052-0.112)	<0.001	0.026 (0.000-0.052)	0.059	0.078 (0.012-0.132)	0.020
Reference model + MD index	0.042 (0.012-0.060)	<0.001	0.024 (-0.002-0.054)	0.079	0.030 (-0.020-0.088)	0.436

Reference model includes age, sex, Townsend scores, ethnicity, smoking status, alcohol intake, hypertension, total energy intake and BMI.

PDI: Overall plant-based diet index; hPDI: healthful plant-based diet index; uPDI: unhealthy plant-based diet index; DASH: Dietary Approaches to Stop Hypertension; MD: Mediterranean diet.

Comment #27: The assessment of each component of the EAT Lancet Diet–CMD association.

Response: Based on your suggestion, we have added a sensitivity analysis assessing each component of the EAT Lancet Diet–CMD associations. We found that in the Stubbendorff EAT-Lancet index, higher adherence to the vegetable recommendation was significantly associated with lower risks of depression, anxiety and their co-occurrence (all *P* for trend <0.05). Similar associations were observed for the other two indexes. In the revised manuscript, we have added these results into the supplementary materials, and details can be found on Page 9, Lines 153-159.

Comment #28: Regarding the co-occurrence, could we have estimates for onset of depression alone (after excluding anxiety) and anxiety alone (after excluding depression) and for onset of any CMD (anxiety or depression)?

Response: Based on your comment, we have estimated the associations of the three EAT-Lancet indexes with onset of depression alone (after excluding anxiety) and anxiety alone (after excluding depression) and with onset of any CMD (anxiety or depression). Higher adherence to the EAT-Lancet diet characterized by the three EAT-Lancet indexes were all associated with lower risks of depression alone, anxiety alone and any CMD, and the results are presented in **Table 5** below.

Table 5 Associations between the EAT-Lancet indexes and risks of depression alone (after excluding anxiety), anxiety alone (after excluding depression) and any CMD (anxiety or depression)

	Depression alone	Anxiety alone	Any CMD
	HR (95% CI)	HR (95% CI)	HR (95% CI)
Knuppel EAT-Lancet index			

≤9	REF	REF	REF
=10	0.901 (0.817-0.993)	0.949 (0.873-1.032)	0.929 (0.875-0.985)
=11	0.894 (0.810-0.986)	0.922 (0.848-1.003)	0.909 (0.856-0.964)
≥12	0.825 (0.734-0.927)	0.836 (0.758-0.921)	0.819 (0.764-0.878)
P for trend	0.0022	0.0003	<0.0001
1-point increment in diet score	0.954 (0.925-0.983)	0.959 (0.934-0.984)	0.953 (0.936-0.971)
Stubbendorff EAT-Lancet index			
≤17	REF	REF	REF
18-20	0.916 (0.817-1.028)	0.890 (0.804-0.984)	0.907 (0.846-0.974)
21-23	0.824 (0.736-0.922)	0.852 (0.773-0.939)	0.834 (0.778-0.893)
24-26	0.774 (0.686-0.873)	0.802 (0.723-0.889)	0.793 (0.738-0.853)
≥27	0.726 (0.627-0.841)	0.794 (0.704-0.896)	0.751 (0.689-0.820)
P for trend	<0.0001	<0.0001	<0.0001
1-point increment in diet score	0.975 (0.966-0.985)	0.984 (0.976-0.992)	0.979 (0.974-0.985)
Kesse-Guyot EAT-Lancet index			
Quintile 1	REF	REF	REF
Quintile 2	0.863 (0.776-0.959)	0.914 (0.835-1.001)	0.899 (0.844-0.958)
Quintile 3	0.839 (0.753-0.934)	0.886 (0.809-0.970)	0.858 (0.805-0.916)
Quintile 4	0.852 (0.765-0.950)	0.850 (0.776-0.932)	0.856 (0.802-0.913)
Quintile 5	0.855 (0.766-0.955)	0.839 (0.765-0.921)	0.840 (0.786-0.898)
P for trend	0.0078	<0.0001	<0.0001
100-point increment in diet score	0.870 (0.789-0.960)	0.847 (0.781-0.919)	0.854 (0.805-0.905)

Models were adjusted for age, sex, Townsend scores, ethnicity, smoking status, alcohol intake, physical activity, hypertension, total energy intake and BMI.

RESULTS

Comment #29: L 175: please add indications on co-occurrence of depression and anxiety.

Response: Thank you for your suggestion. The co-occurrence of depression and anxiety was defined as developing both depression and anxiety during follow-up, which has been clarified in the revised outcome identification section (Page 20, Lines 385-387).

Comment #30: Analyses of associations between individual components and CMD would be relevant.

Response: Thank you for your comment. We have added a sensitivity analysis assessing each component of the EAT-Lancet diet–CMD associations and we found that in the Stubbendorff EAT-Lancet index, higher adherence to the vegetable recommendation was significantly associated with lower risks of depression, anxiety and their co-occurrence (all *P* for trend <0.05). Similar associations were observed for the other two indexes. Details can be found on Page 9, Lines 153-159.

Comment #31: Above I suggested few additional analyses, I encourage authors to display the results.

Response: Thank you for your kind suggestions. We have performed all the analyses you mentioned above and added them to our revised manuscript, all of which have helped us to enrich and improve our research. Thank you again!

DISCUSSION

Comment #32: The discussion regarding the novelty and validation of EAT-Lancet diet should be enriched.

I think authors should specify how the EAT –Lancet has been built (or specify this in Intro), on which bases, its targets. i.e. While the EAT-Lancet diet provides a set of recommendations for feeding the entire world population within planetary boundaries, it has been designed to serve as an anchor for integrating sustainability into national dietary recommendations of culturally diverse countries.

Response: Thank you for your comments. Besides adding descriptions about how the

EAT–Lancet has been built and its targets in the introduction section (Page 4, Lines 55-64), we also added discussion about its unique benefits for the environment in the context of the serious negative effects of the current food production on the global environmental health. In addition, we pointed out that the EAT-Lancet diet has been designed to serve as an anchor for integrating sustainability into national dietary recommendations of culturally diverse countries. Details can be found on Page 11, Lines 200-211.

Comment #33: We expect some discussion regarding components, the cut-offs used, which ones seem adapted to mental health outcomes and why? some additional comparison with similar plant-based diet and depression association previously published is lacking. Please refer to the ref I mentioned above (p.2)

Response: Thank you for your comments. Associations of individual components and incident mental outcomes have been discussed in our revised manuscript, including discussion about individual components, the different cut-offs used and scoring criteria among the three EAT-Lancet indexes. Details can be found on Page 12, Lines 216-231. We have also referred to the references you mentioned above and added additional comparisons with established plant-based diet about their predictive abilities of incident depression, anxiety and co-occurrence of depression and anxiety, which can be found on Page 10, Lines 182-189 and Pages 10-11 Lines 193-199.

Comment #34: In addition, line 278, authors mentioned publications assessing EAT

lancet diet and chronic diseases. Authors mentioned studies reporting significant associations, what about those failing to observe any association between EAT Lancet diet and mortality (Knuppel et al.). What do authors think about this following paper? Zgmutt FJ, Pouzou JG, Costard S The EAT-Lancet Commission's dietary composition may not prevent non communicable disease mortality J Nutr., 150 (5) (2020), pp. 985-988.

Response: Thank you for your comments. In the study by Knuppel et al., they found that although the beneficial association of the EAT-Lancet diet with all-cause mortality was statistically significant, but it was not as apparent and robust as those with ischaemic heart disease and diabetes, thus concluding no clear association with all-cause mortality. Previous studies have also observed significant inverse associations between adhering to the EAT-Lancet diet and mortality in Swedish[12], Chinese[13] and UK[14] adults. The Knuppel EAT-Lancet index, built on binary assessments of each diet component, leads to a narrower possible range, and different levels of adherence to the proposed intake ranges were therefore hard to be comprehensively examined, which might affect the association to a certain degree. Differences in the study populations and follow-up time might also contribute to some differences in the associations of adherence to the EAT-Lancet diet with mortality in different studies.

We agree with Zgmutt et al. that transparency and replicability do count, and future validations are needed, especially in different regions of different sociodemographic backgrounds. However, the interpretation of the EAT-Lancet diet is developing and evolving on an ongoing basis, and several methods of translating the EAT-Lancet diet

into diet index have been developed and are continuously coming out. Due to the different constructions between different scoring criteria, studies investigating the associations of adherence to the EAT-Lancet diet with health-related outcomes may preferably use multiple scores to assess the robustness of the associations.

Comment #35: Authors dedicated a section considering potential mechanisms. Authors mentioned inflammation and gut microbiome as stated in all papers reporting research in nutritional psychiatry. By deriving other diet indices, authors would be able to provide how the EAT Lancet diet scores are correlated with Med Diet, DASH, AHEI, Inflammatory diet scores. At least this should be put as perspective. Given the richness of the cohort, we would expect more discussion regarding who benefit the most of adhering to EAT Lancet diet in regards to CMD.

Response: Thank you for your comments. We agree with you that investigating how the EAT-Lancet diet scores are correlated with other well established and widely used diet scores will definitely provide a more in-depth insight into the potential mechanism, and we have put this as a perspective in our revised manuscript, which can be found on Page 14, Lines 263-267.

According to the results of our subgroup analyses (Page 9, Lines 166-170), individuals with different sex and smoking status, to a similar extent, benefit from adhering to the EAT-Lancet diet. However, we found the associations of adhering to the EAT-Lancet index with incident depression and anxiety were more pronounced in those who were more deprived (defined by higher Townsend deprivation index) (all *P* for interaction

<0.05). Discussion can be found on Page 13, Lines 243-249.

Comment #36: What about the mediating role of BMI, CVD, hypertension or T2D?

Does adherence in EAT lancet diet different according to ethnic group all these elements would definitely contribute to improve the current paper.

Response: We observed that about 20% and 8% of the associations between the EAT-Lancet diet and depression and anxiety were mediated by BMI, respectively. The mediating effects of CVD, T2D and hypertension were relatively weak or not significant. Details can be found in Response to Comment #5 above.

Since over 95% of participants in our study are White, we repeated the main analysis in White and observed similar results. We have also discussed the generalizability of our current findings to other populations in our revised manuscript, which can be found on Pages 15-16, Lines 298-302.

References

- [1] COVID-19 Mental Disorders Collaborators. Global prevalence and burden of depressive and anxiety disorders in 204 countries and territories in 2020 due to the COVID-19 pandemic. *Lancet*. 2021;398(10312):1700-1712.
- [2] Salari N, Hosseinian-Far A, Jalali R, et al. Prevalence of stress, anxiety, depression among the general population during the COVID-19 pandemic: a systematic review and meta-analysis. *Global Health*. 2020;16(1):57.
- [3] Molendijk M, Molero P, Ortuño Sánchez-Pedreño F, Van der Does W, Angel Martínez-González M. Diet quality and depression risk: A systematic review and

- dose-response meta-analysis of prospective studies. *J Affect Disord.* 2018;226:346-354.
- [4] Kris-Etherton PM, Petersen KS, Hibbeln JR, et al. Nutrition and behavioral health disorders: depression and anxiety. *Nutr Rev.* 2021;79(3):247-260.
- [5] Sarris J, Logan AC, Akbaraly TN, et al. International Society for Nutritional Psychiatry Research consensus position statement: nutritional medicine in modern psychiatry. *World Psychiatry.* 2015;14(3):370-371.
- [6] Quirk SE, Williams LJ, O'Neil A, et al. The association between diet quality, dietary patterns and depression in adults: a systematic review. *BMC Psychiatry.* 2013;13:175.
- [7] Li Y, Lv MR, Wei YJ, et al. Dietary patterns and depression risk: A meta-analysis. *Psychiatry Res.* 2017;253:373-382.
- [8] Yin W, Löf M, Chen R, Hultman CM, Fang F, Sandin S. Mediterranean diet and depression: a population-based cohort study. *Int J Behav Nutr Phys Act.* 2021;18(1):153.
- [9] Khayyat-zadeh SS, Mehramiz M, Mirmousavi SJ, et al. Adherence to a Dash-style diet in relation to depression and aggression in adolescent girls. *Psychiatry Res.* 2018;259:104-109.
- [10] Wu H, Gu Y, Meng G, et al. Quality of plant-based diet and the risk of dementia and depression among middle-aged and older population. *Age Ageing.* 2023;52(5):afad070.

- [11] Lee MF, Eather R, Best T. Plant-based dietary quality and depressive symptoms in Australian vegans and vegetarians: a cross-sectional study. *BMJ Nutr Prev Health*. 2021;4(2):479-486.
- [12] Stubbendorff A, Sonestedt E, Ramne S, Drake I, Hallström E, Ericson U. Development of an EAT-Lancet index and its relation to mortality in a Swedish population. *Am J Clin Nutr*. 2022;115(3):705-716.
- [13] Ye YX, Geng TT, Zhou YF, et al. Adherence to a Planetary Health Diet, Environmental Impacts, and Mortality in Chinese Adults. *JAMA Netw Open*. 2023;6(10):e2339468.
- [14] Karavasiloglou N, Thompson AS, Pestoni G, et al. Adherence to the EAT-Lancet reference diet is associated with a reduced risk of incident cancer and all-cause mortality in UK adults. *One Earth*. 2023;6(12):1726-1734.

Reviewers' Comments:

Reviewer #2:

Remarks to the Author:

The authors did address all the comments from my previous review. However they did perform an additional analysis that I think does not add anything to the study and it is hard to justify:

"Moreover, we performed additional analysis where we randomly selected 10% of the individuals with incident depression, anxiety or the co-occurrence of depression and anxiety (case group), respectively, and recategorized them into the control group to simulate the underdiagnosed conditions. " - top of page 15.

Since this is a population with already underdiagnosed conditions, this "trial simulation" does not add value to the study. I suggest to omit it from the manuscript and supplementary tables.

Reviewer #3:

Remarks to the Author:

Authors replied to all requests formulated in my review. I do not have additional questions or suggestions.

Reviewer #2 (Remarks to the Author):

Comment #1: The authors did address all the comments from my previous review. However, they did perform an additional analysis that I think does not add anything to the study and it is hard to justify:

"Moreover, we performed additional analysis where we randomly selected 10% of the individuals with incident depression, anxiety or the co-occurrence of depression and anxiety (case group), respectively, and recategorized them into the control group to simulate the underdiagnosed conditions. " - top of page 15.

Since this is a population with already underdiagnosed conditions, this "trial simulation" does not add value to the study. I suggest to omit it from the manuscript and supplementary table.

Response: Thank you for your suggestion. We have omitted the analysis from our manuscript.

Thank you again for all your comments and professional advice that have helped us improve the quality of our manuscript.